



# Aquifer recharge in the Piedmont Alpine zone: Historical trends and future scenarios

Elisa Brussolo[1], Elisa Palazzi[2], Jost von Hardenberg[3,2], Giulio Masetti[4], Gianna Vivaldo[4], Maurizio Previati[5], Davide Canone[5], Davide Gisolo[5], Ivan Bevilacqua[5], Antonello Provenzale[4], and Stefano Ferraris[5]

[1]Research Center, Società Metropolitana Acque Torino S.p.A., Torino, Italy
[2]Institute of Atmospheric Sciences and Climate, National Research Council of Italy (CNR), Torino, Italy
[3]Department of Environment, Land and Infrastructure Engineering (DIATI), Politecnico di Torino, Torino, Italy
[4]Institute of Geosciences and Earth Resources, National Research Council of Italy (CNR), Pisa, Italy
[5]Interuniversity Department of Regional and Urban Studies and Planning (DIST), Politecnico di Torino and Università di Torino, Torino, Italy

**Correspondence:** Elisa Palazzi (e.palazzi@isac.cnr.it)

**Abstract.** The spatial and temporal variability of temperature, precipitation, actual evapotranspiration and of the related water balance components, as well as their responses to anthropogenic climate change, provide fundamental information for an effective management of water resources.

In this study we evaluated the past, present and future quantity of water potentially available for drinking supply in the water

catchments feeding the about 2.3 million inhabitants of the Greater Turin metropolitan area (North-Western Italy), considering climatologies at the quarterly and yearly timescales. Observed temperature and precipitation data from 1959 to 2017 were analyzed to assess historical trends, their significance and the possible cross-correlations between the water balance components. Regional climate model (RCM) simulations from a small ensemble were analysed to provide mid-century projections of the difference between precipitation and actual evapotranspiration for the area of interest under two emission scenarios.

The analysis over the historical period indicated that the driest area of the study region displayed negative annual (and spring quarter) trends of the difference between precipitation and actual evapotranspiration. Close-by relatively wetter watersheds exhibit different behaviors. The analysis of future projections suggested almost stationary conditions for annual data. Regarding quarterly data, a slight decrease in the second half of the year was obtained. The future trends of drainage are very different across the considered RCM ensemble.

The large interannual variability of precipitation, rather than trends, was therefore identified as the most relevant risk factor for water management, expected to play a major role also in future decades.





# 1 Introduction

Water is a crucial resource, intrinsically linked to society and culture development, food and energy security, well-being, envi-
ronmental sustainability and poverty reduction. However, several factors, including urbanization, population growth, land use
and soil consumption, industrial and agricultural development, endanger water resource sustainability in terms of availability,
quality, management and demand (IPCC, 2014; WWAP, 2015). Groundwater resources represent about 97% of liquid fresh-
water resources on Earth (WHO, 2006; Healy, 2010) and play a key role in water supply and proper ecosystems preservation
(WWAP, 2015). Groundwater resources help to maintain river discharges and, together with surface freshwaters, are accounted
for in water budget considerations at the river basin scale (Rumsey et al., 2015). Groundwater resources are of utmost impor-
tance for their mitigation effects during dry periods and their reduction can impact the whole hydrological cycle. Groundwater
is a fundamental natural resource that acts as a reservoir from which good quality water can be collected for drinking purposes,
requiring few purifying treatments compared to surface water.

Climate change influences several components of the water cycle, including groundwater resources, causing a lowering of
piezometric levels due to discharge modifications as a result of snow retention reduction, changes in precipitation regimes
and potential evapotranspiration increase. Surface water and pollutants infiltration together with over-exploitation of wells can
further deplete groundwater resources, triggering the competition between irrigation and potable uses. Though the degradation
of water quality mostly depends on land use and saltwater intrusions into coastal groundwater (Jiménez Cisneros et al., 2014),
the availability and the quality of groundwater are strictly linked to each other.

Climate change also exacerbates the risks associated with changes in the distribution and availability of water resources
(Jiménez Cisneros et al., 2014), with consequences on water-demand management and infrastructural system planning. In this
framework, assessing climate change impacts on Integrated Urban Water Management, considering a worsening of preexisting
conditions and/or an occurrence of new hazards or risk factors, and planning climate change adaptation strategies are funda-
mental challenges that Integrated Urban Water Management is expected to face in the near future, using an integrated approach
based on prevention, preparedness and risk assessment.

The Alps and the Mediterranean area are recognised as two climate hot-spot regions (IPCC, 2014), showing amplified
climate change signals and associated environmental, social and economical impacts. Future projections for the Italian territory,
in particular, show an increase of high precipitation intensity typically distributed in more intermittent events, together with
an increase of the duration of dry periods (Desiato et al., 2015). However, in northwestern Italy these trends are not clear and
they are very site-dependent (Baiamonte et al., 2019). Italy is a climatic bridge between the Mediterranean and the inland
European climate (Libertino et al., 2019) and in Piedmont there are 4000-meter-high mountains at just 160 km away from the
Mediterranean sea.

Northern Italy is also a bridge between areas where actual evapotranspiration is mainly soil-moisture limited (Mediterranean)
and areas where it is energy-limited (Central Europe). The widespread use of irrigation in NW Italy is decreasing the soil
moisture limitation; in the mountain portions, the wide forests are mainly energy limited. However, mountain grasslands and
not-irrigated areas, where moisture limitation is present, still play an important role.



For precipitation, the transient development of the impingement of cold fronts on the Alps induces a wide range of mesoscale phenomena. On one hand, when the Alpine chain is subject to a southerly flow of moist and relatively warm air from the Mediterranean Sea, very intense precipitation episodes can take place, such as the Piedmont flood in November 1994. On the
other hand, northerly flows lead to dry weather (Pradier et al., 2002). The main part of relevant rainfall and snowfall episodes, including extreme events, are thus due to southerly flow.

For water management, the Integrated Urban Water Management in Italy is geographically organized into local districts (called "ATO"– Ambiti Territoriali Ottimali, which translates into "Optimal Territorial Divisions") whose domains were defined based on various criteria including river basin boundaries (Legislative Decree No 152/2006, as further amended). The
boundaries of these districts mostly coincide with administrative borders; in the Piedmont region, the Greater Turin metropolitan area represents local district ATO3, where the Integrated Water Management Services are provided by Società Metropolitana Acque Torino (SMAT) S.p.A. This is a wide and geographically complex area and SMAT exploits many and diverse supply sources, with groundwater resources representing about 80% of the whole water supply in terms of volume available to SMAT.

It is therefore important to evaluate the balance between precipitation and actual evaotranspiration (AET), and the related spatial and temporal variabilities. Several studies can be found in the literature which evaluate the impacts of climate change on groundwater resources (Jiménez Cisneros et al., 2014).

Recharge (for the sake of simplicity here defined as the difference between precipitation and actual evapotranspiration ) varies in space and time and it is difficult to measure directly, therefore a comprehensive understanding is lacking. A new
global dataset encompassing more than 5000 locations has shown that precipitation amounts and seasonality of temperature and precipitation are the most important variables. However, also soil and vegetation play an important role (Moeck et al., 2020; Condon et al., 2020).

Regarding spatial variability, Pangle et al. (2014) identified drainage as a proxy of recharge in a controlled mesocosm. Their results highlighted the potential for local interactions between temperature, vegetation, and soils to moderate the hydrologi-
cal response to climate warming. Their AET decreased in summer because of soil moisture shortage. The local climate had precipitation out of phase with growing season. Also, they did not find a reduction of AET at the yearly scale.

Regarding temporal variability, interannual variability also plays a major role in groundwater recharge, and it is critical for water managers. Masbruch et al. (2016) has shown how quasi-decadal large groundwater recharge events can be important for replenishing the aquifers. These events are characterised by large precipitation (both rainfall and snow water equivalent) and
by below-average seasonal temperatures.

Also, the effects of climate change are not all in the same direction. In a study using 16 GCMs, a considerable uncertainty in both the magnitude and direction of recharge changes was shown for 2050 year projections in different parts of the High Plains (Crosbie et al., 2013). A more recent study has revealed variability in both direction and magnitude of hydrological changes for the Great Lake basin of North America, with a combination of different RCMs (Persaud et al., 2020).

Konapala et al. (2020) reported at the global scale an increases in annual mean evaporation over the land surface, attributed to the increase in temperature. They deal with water availability as the difference between precipitation and actual evapotran-





spiration. In this paper we call it drainage, as a proxy of recharge, computed by the soil model in each pixel, without modeling both runoff and the underlying aquifer flow.

In this study we focused on an area where the aquifer providing water to about 2.3 million people is catched from watersheds

characterized by a very large spatial precipitation variability, owing to the proximity of high mountains and of the sea.

We evaluated the temporal variability and the trends of the water balance terms, estimating the quantity of groundwater resource available for drinking water supply in the water catchments of the area managed by SMAT. The analyses are performed at both the quarterly and the hydrological year timescale, for both past and future conditions, analyzing a historical climate dataset for the period 1959–2017 and future projections from regional climate model simulations up to 2050, in order to be

compliant with relatively short-term water management objectives.

Given the timescales of interest and the uncertainties inherent in some terms of the water balance equation, in this study the standard formulation of this equation has been simplified, including precipitation, actual evapotranspiration (AET), and drainage (obtained by subtracting AET from the sum of rainfall and snowmelt) which is used as a proxy of the groundwater recharge. This approach is common to other studies and many of them (Healy, 2010) do not make a distinction between

drainage and groundwater recharge. In the study area measured flow data cover a temporal interval shorter than two decades (2000-2017), not allowing to build robust regressive models able to estimate deep percolation outside the time interval of data availability and drainage is the only water balance variable which can be evaluated by using the meteorological variables as inputs.

The objective of this study is to evaluate the future change and variability of precipitation, AET, and drainage, in order

to obtain future projections of aquifer recharge for the area of interest, at the hydrological year and quarters timescales. The expected result will then form a knowledge basis for operational indications. This study represents a scientific contribution to the management and the governance of water resources and water supply, through e.g. the implementation of scientifically-driven guidelines and strategic agendas on water supply and water policies.

The paper is structured as follows: Section 2 describes the study area and the water balance terms at the catchment scale;

Section 3 describes the employed meteoclimatic and hydrological data; Section 4 deals with the soil-water model and its characteristics and use in the present study; Section 5 is about future projections obtained with the regional climate models employed in this study and the considered climate variables; Section 6 presents the results and Section 7 discusses them and concludes the paper.

## 2   Study area and methodology

### 2.1   Study area

Many foothill zones in the Alps and Apennines contain aquifer systems of strategic interest for water supply, especially for drinking purposes (Doveri et al., 2016). In this framework the aquifer system extending in the foothill plain located in the Piedmont region between the western Alps and the Turin hills is extremely relevant. This study area, within the administrative borders of the Greater Turin metropolitan area, has a complex orography and is surrounded on the western and northern sides



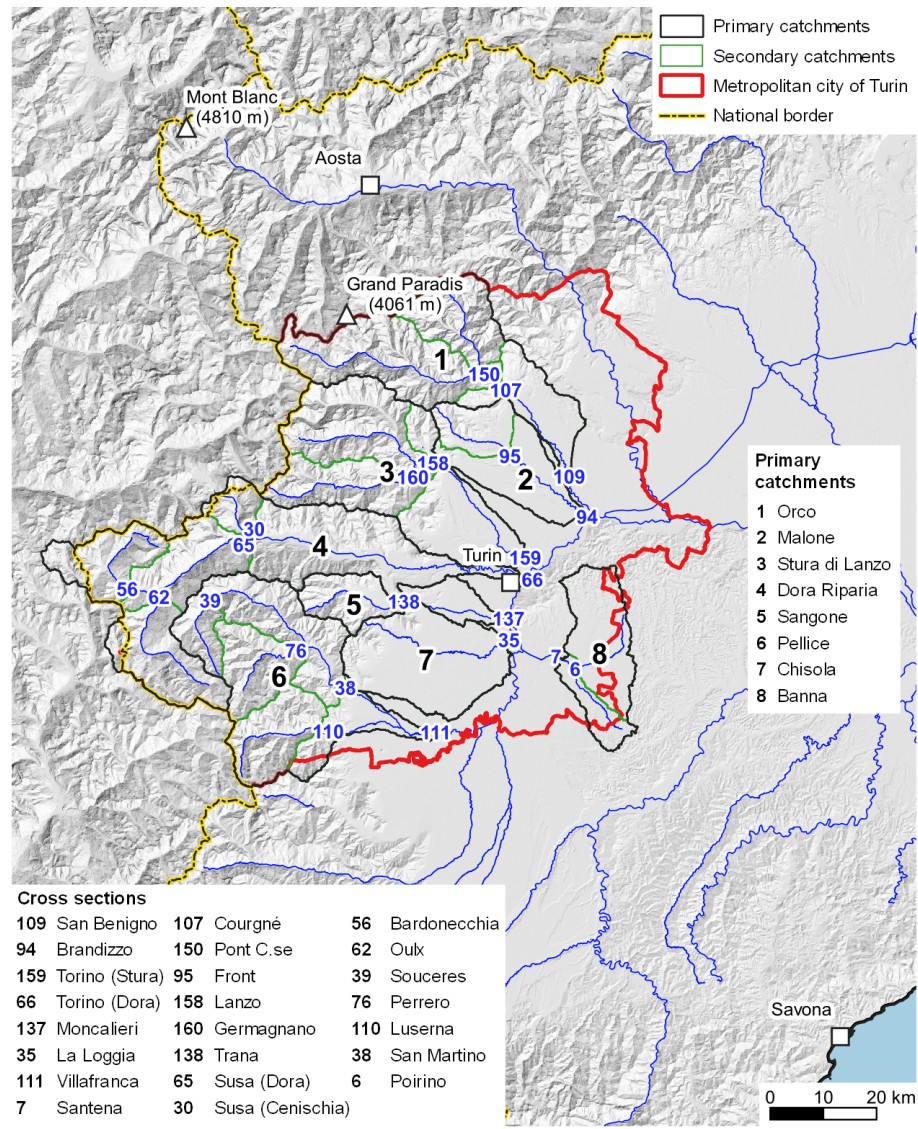

**Figure 1.** The study area including river catchments and sub-catchments. Topographic shading is based on DEM data from the Progetto Risknat - Base topografica transfrontaliera, ARPA Piemonte (http://webgis.arpa.piemonte.it/ags101free/rest/services/topografia_dati_di_base/Sfumo_Europa_WM/MapServer).

by the Alps (with elevation peaks higher than 4000 meters above sea level at the border with the Valle d'Aosta region) and on the eastern and southern sides by hills and plains. Precipitation in the study area is characterized by relatively high spatial and temporal variability and affected by both local and large-scale circulations. The study area, in fact, is prone to topographically-induced precipitation and is exposed to the inflow of moisture-rich air from the Mediterranean sea (Ciccarelli et al., 2008). It is also an area characterized by the occurrence of relatively long dry periods (Agnese et al., 2012; Baiamonte et al., 2019).





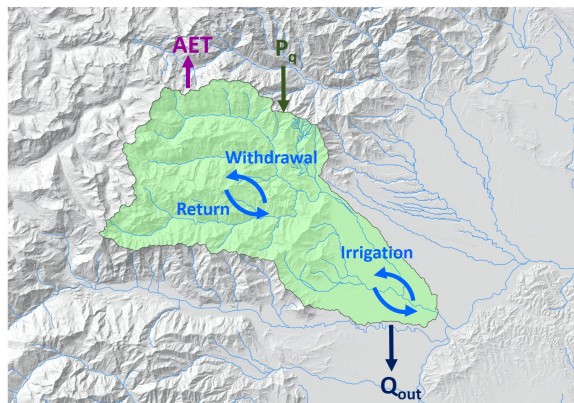

**Figure 2.** Illustration of the water balance terms at the catchment scale. Topographic shading is based on DEM data from the Progetto Risknat - Base topografica transfrontaliera, ARPA Piemonte (http://webgis.arpa.piemonte.it/ags101free/rest/services/topografia_dati_di_base/Sfumo_Europa_WM/MapServer).

Owing to its hydrogeological features (see also the rivers system shown in Fig. 1), the foothill aquifer systems generally sustain the infiltration of both local rainfall and streamwater originating in mountain catchments. These systems are highly sensitive to variation and changes in meteoclimatic variables, first of all in precipitation regimes.

## 2.2 Water balance terms

In this paper we refer to water balance as the balance between water inputs and outputs at the catchment scale, as schematically shown in Figure 2.

Being precipitation much more episodic than evapotranspiration, daily time steps are recommended with respect to longer ones, in order to avoid the underestimation of drainage and recharge (being the distinction between them one of lag time (e.g. Healy, 2010). Therefore in this work the computation is done at the daily scale in the soil column, and results are then aggregated at the yearly and quarterly time scale and, spatially, at the catchment scale.

The daily soil model will be described later in the text. The yearly catchment water balance can be simplified as follows (Healy, 2010):

$$P_q = AET + Q_{out} \tag{1}$$

where $P_q$ represents the sum of liquid precipitation (rainfall) and snowmelt, $AET$ the Actual Evapotranspiration and $Q_{out}$ the drainage. Input surface or subsurface flow can be neglected because all catchments are bordered by the mountain divide. Storage variation in time is disregarded, because the control volume is composed of vadose zone soil columns, with an aggregation of results at the quarterly and yearly time scale. The soil model presented in the following (Sec. 4) is used to calculate $AET$ and $Q_{out}$ in each pixel at the daily time scale.





To account for the different ground characteristics and variations, together with the physical description of the hydrological processes, all the water balance variables (except for output discharge) must be first evaluated at a fine spatial resolution and then aggregated (i.e., upscaled) at watershed level. To this end, a horizontal spatial resolution of 250 m was considered as

a good compromise between the necessity of high-resolution and the computational resources required: the study area was discretised into a grid of 652 x 521 pixels, with a 250 m $\times$ 250 m spatial resolution.

The model takes into account the effects of irrigation on the value of $AET$. However, input and output irrigation terms in surface and groundwater balance are assumed to balance out at the catchment level. Industrial water withdrawals are mainly for hydroelectric production and therefore they are not changing the water balance at the watershed scale, because they give

back the captured water at short distance.

The contribution of glacier and permanent snow melting is disregardable with respect to the catchment scale. However the evaluation of snowmelt from the snowpack is quite important as it represents the amount of daily snow equivalent contributing to the water balance. To this end, as a first step, air temperature was used to distinguish between snowfall ($P_s$) and rainfall ($P_r$) (DeWalle and A.Rango, 2008): for temperatures lower or equal to -2° C all the precipitation amount can be considered

as snowfall, while for temperature higher or equal to 5° C all the precipitation amount can be considered as rainfall. For temperatures between -2° C and 5° C, we linearly interpolated rainfall and snowfall amounts to breakdown the different percentages of precipitation type. The resulting snowfall $P_s$ is used as an input to a a simple bucket model in each pixel, used to integrate the storage of snow on ground, $S$. The daily snowmelt, $N_f$, is subtracted from the same bucket and it is evaluated using the method by Zeinivand and Smedt (2009), where the melted snow is a function of the difference between the daily

mean temperature $T_{mean}$ and melting temperature $T_0$ (0° C) and of the rainfall amount ($P_r$ evaluated in mm/day), as follows:

$$N_f = \min\left\{ S/(1\text{ day}), \max\left[0, (k_{snow} + k_{rain} \cdot P_r) \cdot (T_{mean} - T_0)\right]\right\} \tag{2}$$

were $k_{rain}$ (rainfall melt-rate factor) and $k_{snow}$ (melt-rate factor) are fixed parameters. We use the same values as in Zeinivand and Smedt (2009): $k_{rain} = 0.0757\ ^\circ\text{C}^{-1}$ and $k_{snow}$=3 mm day$^{-1}\ ^\circ\text{C}^{-1}$ .

A correct evaluation of the water balance allows a reliable estimation of water availability. To this end we applied the following chain in the study area:

1. identification and retrieval of the daily meteoclimatic (i.e. temperature and precipitation) data;

      2. regridding of the meteoclimatic and hydrological data at the spatial resolution required by the hydrological model, namely a square 250 m grid;

      3. setting-up of a mathematical model that accounts for soil-vegetation-atmosphere interactions and providing actual evapotranspiration estimates (see Section 4).

Observational and reanalysis datasets have been used to obtain meteoclimatic data for the period 1959-2017, as described in Section 3, while for the assessment of future water balance scenarios we relied on climate projections of temperature and precipitation (see Section 5).



In order to be compliant with management objectives for strategic long-term planning, the input datasets, available at daily resolution (see Section 3), were aggregated to quarterly and yearly timescales using the definition of hydrological year (the first

quarter includes January, February and March ($JFM$), the second April, May and June ($AMJ$), the third July, August and September ($JAS$) and the fourth October, November and December ($OND$)). The hydrological year $N$, starts on October 1st of year $N-1$ and ends on September 30th of year $N$. During the hydrological year a complete snowfall melting occurs, making the breakdown of precipitation between rainfall and melted snow negligible. At high altitudes this melting is not complete and the annual variability entails the fluctuation of snow at soil from one (hydrological) year to the following one. This phenomenon

is limited to the highest peaks and is relevant only for limited areas. For this reason for evaluations at the yearly timescale $P_q$, was assumed to coincide directly with total average yearly precipitation.

## 3   Observed data

The past and present precipitation and temperature data necessary to drive the hydrological model employed in this study derived from the Regional Environmental Protection Agency of Piedmont (ARPA Piemonte) databases. In particular they

have been extracted from the OI dataset (in italian, ARPA Piemonte, 2010a; Ciccarelli et al., 2008). This dataset provides cumulative daily precipitation, and maximum and minimum daily temperature at a spatial resolution of $0.125°$ longitude-latitude (corresponding to $\sim$12 km) over the entire Piedmont region, for a time period extending from 1959 to 2017. The OI dataset was obtained by interpolation of in-situ observations collected by the Hydrographic Office network and by the network of the ARPA telemetry stations through the technique of the Optimal Interpolation, which allows to obtain data on a

regular grid homogenizing observational data from different measurement networks and sources (ARPA Piemonte, 2010b). A preliminary quality check of the OI data revealed the existence of days (all referring to years before 1990) where the minimum temperature showed a higher value than the maximum temperature, probably owing to issues in the data acquisition. These data were excluded from the analysis and replaced by new values obtained through linearly interpolation in time, rather than in space, in order not to destroy the orographic information inherent in the original OI dataset.

The OI dataset provides a set of meteorological variables at a coarser spatial resolution than that required to describe the small scale hydrological processes and to accurately estmate the water balance terms, calling for the application of interpolation and downscaling techniques. After re-projecting the OI temperature and precipitation data into a WGS84/UTM 32N coordinate system useful for the subsequent analyses, they were further interpolated at the finer resolution of 250 m over the domain of interest ($x_{min} = 304250W$, $x_{max} = 434500W$, $y_{min} = 4951750N$, $y_{max} = 5094750N$). The conversion was preceded by a

preliminary bilinear interpolation of the OI dataset to produce an intermediate higher resolution dataset at a resolution of 0.001 ∘ latitude-longitude, to avoid artifacts when re-projecting into UTM coordinates. Post-processing was performed using the GDAL (Geospatial Data Abstraction Library; v. 2.1) and CDO (Climate Data Operators; v. 1.7.0) software.

A further adjustment to account for orographic effects was applied to maximum and minimum temperature data, using the environmental lapse-rate correction coefficients derived for the Alpine region by Rolland (2003) shown in Table 1. Figure 3

shows an example of orographic correction (middle panel) applied to a maximum temperature field (left panel) for a selected





**Table 1.** Average climatological monthly adiabatic lapse-rate from Rolland (2003) for Alpine region (expressed in $K/km$)

| Jan | Feb | Mar | Apr | May | June | July | Aug | Sep | Oct | Nov | Dec |
|-----|-----|-----|-----|-----|------|------|-----|-----|-----|-----|-----|
| 4.5 | 5.0 | 5.8 | 6.2 | 6.5 | 6.5  | 6.5  | 6.5 | 6.0 | 5.5 | 5.0 | 4.5 |

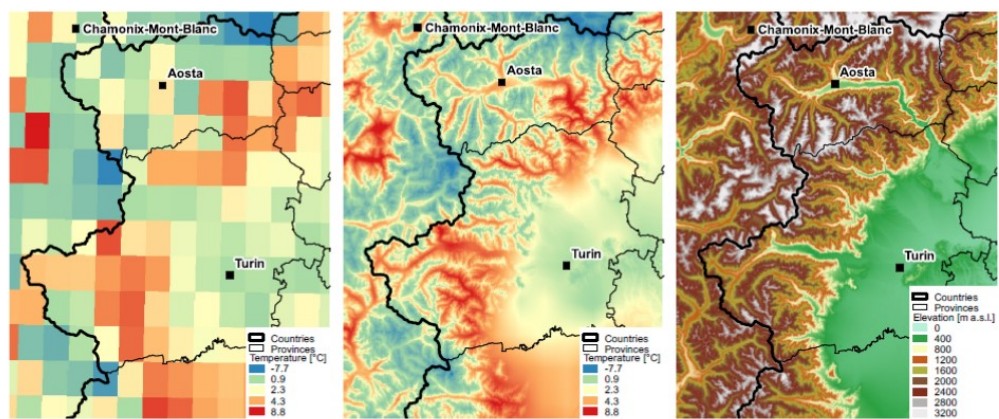

**Figure 3.** Maximum temperature field at (left) 0.125° lon-lat spatial resolution from the original OI dataset and at (middle) 250 m resolution after interpolation with orographic correction, exemplary for January 1st, 2009. The right panel shows the orography of the study area from a Digital Elevation Model derived from the SRTM project, interpolated at 250 m of spatial resolution.

day (January 1st, 2009). The orography of the study area is shown in the right panel, based on a 90 m resolution Digital Elevation Model derived from the SRTM (Shuttle Radar Topography Mission) project (Jarvis et al., 2008), interpolated at 250 m.

## 4   Soil-water model

A simple bucket soil-water model was developed in order to estimate AET values at the daily scale for each 250 m × 250 m pixel of the study area. The soil is schematised with seven different layers of increasing thickness with depth, similar to the FAO56 model (Allen et al., 1998). A simplified scheme is shown in Figure 4, which includes only one layer of the bucket model. The water inputs for the model are precipitation (sum of rainfall and melted snow) and irrigation, if any. The outputs are AET and drainage (sum of runoff and deep percolation). The distinction between runoff and deep percolation is not included in the model, as well as the modelling of capillary rise.

For each layer, the bucket equation is as follows:

$$P_q + I^* - Q_{out} - AET - \Delta S = 0 \tag{3}$$





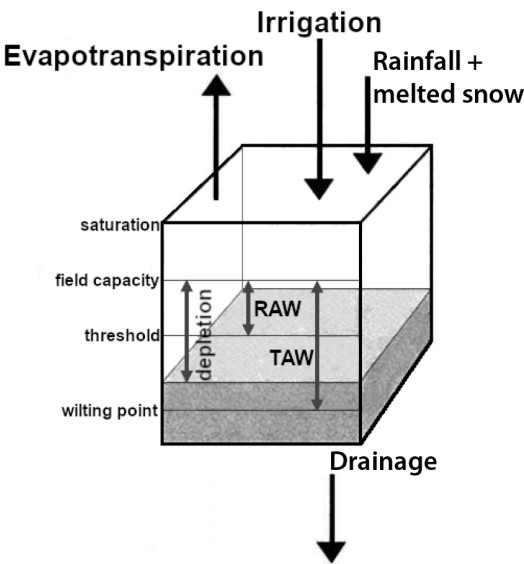

**Figure 4.** Schematic functioning of the bucket model, with only one single layer shown. Adapted from Allen et al. (1998).

where $P_q$ is rainfall plus snowmelt; $I^*$ is irrigation; $Q_{out}$ is drainage; $AET$ is actual evapotranspiration; $\Delta S$ is the variation of water storage in the soil per day. In a bucket model (the one by Baudena et al. (2012) was applied in the same area in northwestern Italy) water can flow upwards within each layer via evapotranspiration or evaporation, and downwards via deep
percolation, as a function of soil water content of each layer: if it exceeds field capacity, the water surplus percolates into the layer immediately below.

### 4.1 Input data of the soil water model

The soil hydraulic properties have been estimated via *pedotransfer* (PTF) following Schaap et al. (2001) from the sand, clay and silt percentages taken from the Soil map of Regione Piemonte (scale 1:250000, IPLA, 2007). Computing via PTF the
wilting point ($WP$) and field capacity ($FC$), the Total Available Water (TAW) is calculated for each layer as follows:

$$TAW = (FC - WP) \cdot (1 - r) \cdot L_d \tag{4}$$

where $r$ is the fraction of volume occupied by stones, and $L_d$ is the layer depth, which is obtained by dividing by seven the root zone depth $z$. The root zone depth has been obtained using the land cover classes from the BDTRE database (Regione Piemonte, 2018), listed in Table 2, converted into root zone depth using different root zone depths for each class, as listed in Table 3. The resulting root zone depth was compared with that also provided by the Regione Piemonte Soil Map (IPLA,
2007), choosing the minimum between the two. For trees the water from the whole soil depth can be depleted and the IPLA depths have been used. Because of the high spatial resolution of BDTRE, the raster has been produced with a 5 m × 5 m grid,





**Table 2.** Land cover classes

| Class | ID Code |
|---|---|
| Winter crops | 1 |
| Summer crops-irrigated | 2 |
| Plain meadows (< 1000m asl) | 3 |
| Orchards | 4 |
| Horticulturals | 5 |
| Plain broadleaves (< 800m asl) | 6 |
| Mountain broadleaves (> 800 m asl) | 7 |
| Coniferous | 8 |
| Vineyards | 9 |
| Mountain grassland (> 1000m asl) | 10 |
| Bare soil | 11 |
| Bare rocks | 12 |
| Impervious surfaces | 13 |
| Water | 14 |
| Others | 15 |
| Glaciers and permanent snow | 16 |

**Table 3.** Soil bucket depth as a function of land cover class. For trees (codes 4,6,7,8,9) the full soil depth from the Soil Map of Regione Piemonte is used.

| ID Code | Soil use | Soil bucket depth [mm] |
|---|---|---|
| 1 2 | Irrigated crops | 1000 |
| 3 5 | Plain grassland and horticultures | 500 |
| 10 | Mountain grassland | 250 |
| 11 | Bare soil | 150 |
| 12 13 14 16 | Impermeable surfaces, water, glaciers | 0 |

then resampled at 250 m by the mode criterium. Homogenising data as a function of the evapotranspiration behavior of each class, the different land cover classes have been aggregated into 16, as reported in Table 2.

Finally, considering that irrigation (here considered as water input) significantly modifies actual evapotranspiration, for
quantifying the actually irrigated fields, we use the Regional Irrigation Information system (Regione Piemonte, 2016), the agricultural crop survey and the historical maps within the actually utilised agricultural areas survey database (Regione Piemonte, 2006), for the years 2013, 2014 and 2015. The actual quantity of irrigation water has been evaluated using the measured data



collected in experiments performed in real world farms both for surface irrigation (Canone et al., 2015) and for sprinklers (Canone et al., 2016).

## 4.2 Actual Evapotranspiration computation

The Actual Evapotranspiration $AET$ was calculated starting from potential evapotranspiration $PET$, then reduced by considering the actual soil water content, obtained from the bucket soil model for each layer, with $AET$ of each day being the sum of the depletions from the single layers. $PET$ was calculated with the model of Hargreaves and Samani, as in Allen et al. (1998), using daily maximum and minimum air temperature data from the OI dataset and extraterrestrial radiation modeled as in Aguilar et al. (2010). Also, the reduction of evapotranspiration (from PET to AET) due to actual soil water content was modeled by using the coefficients Kc and Ks related to crop and soil respectively according to Allen et al. (1998).

## 5 Future projections of precipitation and temperature

Reliable estimates of the hydrological response and of water availability in the coming decades generally require the implementation of a modelling chain consisting of global climate models (GCMs), which provide climate scenarios for the entire planet, regional climate models (RCMs) nested into global models providing lateral and boundary conditions for the regional simulation and, depending on the resolution which needs to be achieved, further downscaling procedures. At the end of the modelling chain, a hydrological model is thus forced with a high-resolution climatic input to simulate the hydrological response at the scale of interest. In this study, we used a small multi-member ensemble of the RCA4 regional climate model forced by five GCM simulations, able to provide the climatic variables of interest – minimum and maximum temperature and precipitation – at a spatial resolution of ∼12.5 km. More detailed analyses of the interplay between several GCMs and RCMs (Sorland et al., 2018) are out of the operational water management scope of this paper. In this work we proceeded similarly to another recharge study (Allen et al., 2010) which used four GCMs and one RCM. They concluded that a range of GCMs should be considered for water management planning. The simulation outputs of this RCM from 1970 to 2050 have been analysed, considering two different emission scenarios for future projections among those defined by the IPCC (Intergovernmental Panel on Climate Change, (IPCC, 2013; Moss et al., 2010)), as described in section 5.1. The model data were subsequently interpolated applying the same procedure employed for the OI observational dataset (see section 3) and adjusted to correct the systematic bias which the model displays with respect to the OI reference climatology in a common time period (1986–2015, see section 5.3 for a description of the post-processing procedure).

## 5.1 RCA4 Regional Climate Model

We analyzed the available simulations of the RCA4 RCM (Strandberg et al., 2014) driven by five different GCMs, namely EC–Earth, CNRM–CM5, IPSL–CM5A–MR, HadGEM2–ES, MPI–ESM–LR, which provide lateral and boundary conditions for the regional simulation. Using one single RCM allowed us to obtain an ensemble of reasonably homogeneous simulations at the regional level, but representing at the same time model uncertainties in future projections captured by the different large-scale





GCMs. RCA4 is one state-of-the-art RCM which participated to CORDEX, the Coordinated Regional Climate Downscaling

Experiment (http://wcrp-cordex.ipsl.jussieu.fr/, Giorgi et al., 2009) sponsored by WRCP (World Climate Research Program) aimed at providing a global coordination of regional climate downscaling activities useful to support climate change adaptation policies. The simulations used in this study, in particular, are part of the EURO-CORDEX initiative (http://www.euro-cordex. net/) which provides regional climate projections for Europe at two different spatial resolutions, ∼50 km (EUR-44, 0.44 degrees resolution) and ∼12 km (EUR-11, 0.11 degrees resolution). EURO-CORDEX includes a total of seven RCMs nested

into several GCMs whose simulations belong to the most recent Climate Model Intercomparison Project phase 5 (CMIP5 Taylor et al., 2012). We chose the RCA4 model because, at the time we downloaded the data from the CORDEX archive, it was the only RCM providing data at the finest available spatial resolution (∼12 km) and with sub-daily (3hr) temporal resolution (WCRP, 2009). For a detailed description of the RCA4 model and its validation, please refer to the technical report by Strandberg et al. (2014).

**5.2 Emission scenarios**

Future projections provided by climate models are based on a set of assumptions about the future evolution of the society in terms of energetic and technological choices, population growth, land use changes, and others, which correspond to possible greenhouse gas emission and concentration pathways in the atmosphere. The latest IPCC report (IPCC, 2013) uses four different Representative Concentration Pathway (RCP) scenarios (Moss et al., 2010; Van Vuuren et al., 2011) to evaluate how

climate is likely to change by the end of the 21$^{st}$ century. For this study we considered two of these scenarios, referred to as RCP4.5 and RCP8.5, as they were the only ones available for the model under consideration. RCP4.5 is a stabilisation scenario in which emissions will be stabilized by 2070 and carbon dioxide concentrations in the atmosphere are expected to stabilise at about twice the pre-industrial level by the end of the century. RCP8.5 is an extreme business-as-usual scenario in which greenhouse gas emissions are not expected to stabilise and carbon dioxide concentrations will be more than tripled at the end

of the century compared to preindustrial levels.

**5.3 RCM data post-processing**

RCA4 simulation outputs were linearly interpolated on the UTM grid at 1 km resolution in the study area using the same method applied to the OI data and illustrated in section 3. To avoid distortions and artifacts, the data were first mapped with the tool CDO (Climate Data Operators; v 1.7) on a lon-lat grid at 0.001° resolution and then projected with the GDALwarp

(Geospatial Data Abstraction Library; v. 2.1) on the final UTM grid (WGS 84 / UTM zone 32N) at 1 km resolution. The intrinsic imperfections of climate model parameterizations and the errors in the model initialization are often reflected in an imperfect representation of the observed climate, which can give rise to biases. Model biases must be taken into account when the climate model outputs are used in impact studies, as impacts and feedbacks can be sensitive to the absolute values and the statistical properties of the climatic input. Bias correction methods are usually applied to correct the differences between climate model

output and observed climatologies and are different depending on the variable and specific application which is considered (Hempel et al., 2013; Maraun, 2013; Maraun et al., 2010). In this study we used an additive correction factor for adjusting the





temperature and a multiplicative correction factor for precipitation (a standard procedure for positive-defined fields) applied pixel by pixel and constant in time, in order to correct the differences in the long-term climatology calculated over a common time period between the simulated and observed fields. To this end, we calculated the long term mean of the historical Euro-CORDEX simulations and of the OI dataset, already interpolated on the UTM grid, in the 30-year long period from 1986 to 2015. In order to maintain the physical consistency between the minimum temperature and the maximum temperature, and thus avoid possible inversions, the same correction factor was used for daily minimum and maximum temperature data, calculated from the average between the daily minimum temperature and the maximum temperatures. The correction factor calculated to correct the model bias in the historical reference period is then applied to the future simulations. In addition to the bias adjustment, temperatures have been further corrected for the lapse rate, as already done for the OI data, based on Rolland (2003).

## 6 Results

### 6.1 Historical trends

Historical data from 1959 to 2017 were analyzed in each river catchment and sub-catchment, aggregating all data previously evaluated at the spatial resolution of 250 m, to find out significant trends both of the water balance terms and the meteorological variables, for both the hydrological year and the quarterly analysis.

A linear regression was performed to calculate the temporal trends. The goodness of line fit was estimated with the coefficient of determination $R^2$ and its statistical significance was evaluated considering the $p-value$ at 5% level (i.e., 95% significance) (Wilks, 2011).

#### 6.1.1 Hydrological year analysis

During the hydrological year, a complete snow melting occurs in the considered area. Thus, in Eq. 1 $P_q$ refers to the whole precipitation amount and $P_q - AET$ to drainage. Figure 5 shows the time series from 1959 to 2017 of the daily maximum temperature (in kelvin, top left), $P_q$ (mm/year, top right), $AET$ (mm/year, bottom left), and drainage (mm/year, bottom right), for the Dora Riparia station in Turin as an example. It is the driest watershed in this study. The west–east oriented valley is in the middle of figure 3c. In the Alps this kind of valley is usually characterised by a very dry and windy weather. Other examples in the Alps are Valais, Valle d'Aosta, Valtellina and Val Venosta.

Figure 5 reveals that, for this catchment, the daily maximum temperature has increased during the study period, with a statistically significant trend. The total precipitation, $P_q$, has decreased (p=0.059) while $AET$ has increased (p=0.054), and drainage has significantly decreased (p=0.032). The catchment considered here is in the driest part of the study region, which is also the part where most of the significant trends of precipitation and AET were detected in the data analysis. The same approach is adopted for all the catchments in the study area. The results are summarised in Table 4 where statistically significant trends are indicated with bold font. For all catchments, hydrological year trends in maximum temperature are positive and



statistically significant, ranging between 0.032 and 0.078 K per year. $AET$ trends are also positive for all catchments and statistically significant in 14 out of 23 cases.

Most catchments exhibit negative precipitation ($P_q$) trends but only 5 cases display statistically significant trends. Precipitation at all catchments exhibit a high interannual variability (not shown here), in keeping with the results of previous studies in the same area (Pavan et al., 2019; Baiamonte et al., 2019; Ciccarelli et al., 2008). Drainage shows negative but mostly non-significant trends in all catchments. The seven catchments with significative decreasing trends all belong to the western and driest part of the region. This area is characterised by much lower yearly total precipitation values than the northern one.

Long-term climatological values of cumulative yearly precipitation in the western Dora Riparia area are: Oulx 552 mm, Susa 710 mm, Beaulard 680 mm, Bardonecchia 724 mm, Pragelato 818 mm, while in the northern part they are: Pont Canavese 1228 mm, Viu 1338 mm, Germagnano 1342 mm. The southern part is characterized by the following values: Cumiana 924 mm, Moncalieri 882 mm.

    To summarise, a slight negative trend is observed for precipitation and drainage over the last 60 years, showing a high spatial

and interannual variability. A quantification of the interannual variability can be obtained evaluating the standard deviation of the detrended timeseries. In Figure 5 the values for the whole Dora Riparia catchment down to Turin are shown. The finding of an increase in AET is not obvious. Pangle et al. (2014) found a decreasing trend of AET looking at data from a mesocosm experiment. Their results highlight that the hydrological response to climate warming can be attenuated where precipitation is seasonal and out of phase with the vegetation growing season. Our results can be due to the irrigated fields in the plain and to

the forested areas in the mountain part of the region, where AET is mostly energy limited. Moreover, in the area of this work the beginning of the growing season is quite rainy. Blyth et al. (2018) also found an increase in evapotraspiration using a land surface model in Great Britain from 1961 to 2015. In a more theoretical work Fatichi and Ivanov (2014) found AET to be quite unaffected by the imposed climate fluctuations, using the input data from four very different sites. The results of this work confirm the spatial variability of the response of water balance to climate change.

### 6.1.2   Quarterly analysis

The meteorological and hydrological variables at the quarterly time scale (January to March, JFM; April to June, AMJ; July to September, JAS; October to December, OND) provide information on intra-annual variations. Figure 6 shows the trend results (color code) for all catchments (identified by their ID number in the y-axis of each panel) and for $T_{max}$ (top left), rainfall plus snowmelt (top right), $AET$ (bottom left) and drainage (bottom right) from 1959 to 2017. The numerical value of the

trend is displayed in a cell when it is statistically significant (p < 0.05). When quarters are considered for the analysis, total precipitation (water input) is defined as the sum of liquid precipitation $P_r$ and snowmelt $N_f$ (see Section 4).

    $T_{max}$ shows positive and statistically significant trends in all catchments and quarters in the period 1959–2017, with values between 0.022 and 0.086 K/year. The increase in the daily maximum temperature is more evident in winter (first quarter) and autumn (fourth quarter). The time series of actual evapotranspiration show positive and significant trends in the majority

of basins and quarters, indicating an overall increase of $AET$ in the entire study area. $AET$ has the greater increase in the

**Figure 5.** Hydrological year time series of $T_{max}$ (top left), $P_q$ (top right), $AET$ (bottom left) and drainage (bottom right) along with their trends, $R^2$, $p-values$ and standard deviations, for the time period from 1959 to 2017, at the Turin cross-section of Dora Riparia river basin (average at the catchment level). The standard deviation, evaluated considering the detrended timeseries, provides a quantification of interannual variability.



**Table 4.** $T_{max}$, $P_q$, $AET$ and $P_q - AET$ trend slope for all the catchments of study area. 1959-2017 time series are considered. The bold font indicates 95% significance.

| Primary catchments | Cross section ID number | $A$ [km²] | $T_{max}$ [K/yr] | $P_q$ [mm/yr/yr] | $AET$ [mm/yr/yr] | $P_q - AET$ [mm/yr/yr] |
|---|---|---|---|---|---|---|
| | 30 | 79.8 | **0.059** | **-2.656** | 0.139 | **-2.795** |
| | 56 | 201.9 | **0.060** | -1.951 | **0.395** | **-2.346** |
| Dora Riparia | 62 | 259.4 | **0.074** | **-3.094** | **0.908** | **-4.002** |
| | 65 | 694.4 | **0.065** | **-2.869** | **0.525** | **-3.394** |
| | 66 | 1242.7 | **0.059** | **-2.909** | **0.292** | **-3.201** |
| | 107 | 628.4 | **0.041** | 0.369 | 0.181 | 0.189 |
| Orco | 109 | 829.3 | **0.042** | 0.155 | 0.235 | -0.080 |
| | 150 | 213.9 | **0.043** | 2.057 | **0.464** | 1.593 |
| | 38 | 581.1 | **0.049** | -1.623 | 0.172 | -1.795 |
| | 39 | 94.8 | **0.078** | **-3.846** | **0.731** | **-4.577** |
| Pellice | 76 | 188.9 | **0.045** | -0.981 | **0.285** | -1.266 |
| | 110 | 213.4 | **0.046** | 0.135 | **0.465** | -0.330 |
| | 111 | 948.2 | **0.048** | -0.991 | **0.277** | -1.268 |
| | 158 | 581.8 | **0.043** | -1.144 | 0.124 | -1.268 |
| Stura di Lanzo | 159 | 898.1 | **0.049** | -1.170 | 0.299 | -1.470 |
| | 160 | 232.3 | **0.044** | -1.954 | 0.111 | -2.065 |
| Banna | 6 | 262.3 | **0.033** | -0.466 | **0.770** | -1.236 |
| | 7 | 347.6 | **0.032** | -0.588 | **0.748** | -1.336 |
| Chisola | 35 | 466.4 | **0.050** | -2.759 | **0.768** | **-3.528** |
| Malone | 94 | 358.2 | **0.054** | -0.920 | **0.927** | -1.847 |
| | 95 | 132.6 | **0.057** | 0.752 | **0.784** | -0.033 |
| Sangone | 137 | 239.5 | **0.053** | -2.897 | 0.128 | -3.025 |
| | 138 | 145.7 | **0.051** | -2.599 | -0.052 | -2.547 |


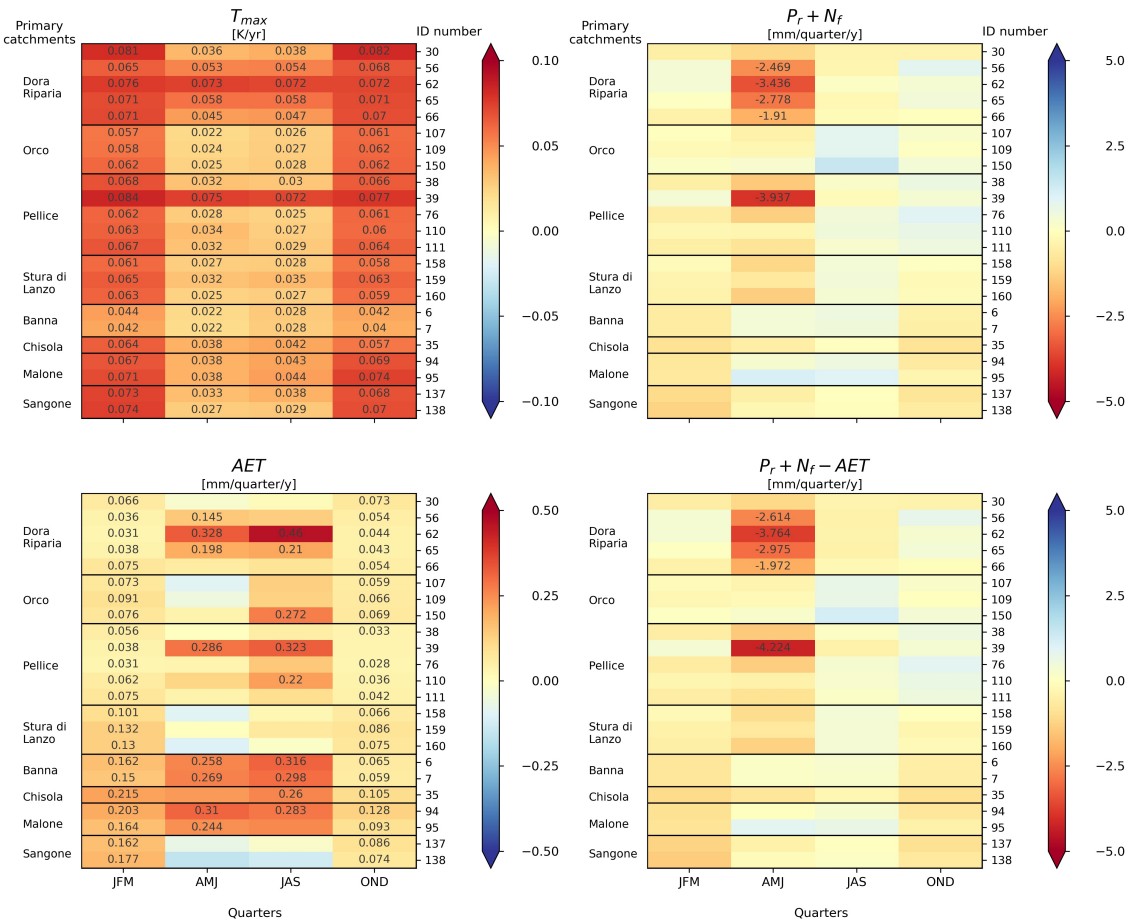

**Figure 6.** Trend slope of all quarters and catchments considered. Top: $T_{max}$ (left) and precipitation (right), bottom: $AET$ (left) and drainage (right). Trend slope is specified if significant at 5% level. When quarters are considered the precipitation is represented by the sum of liquid precipitation $P_r$ and melted snow $N_f$.





summer quarter ($JAS$). Positive trends in the first quarter are consistent with those observed for $T_{max}$, both suggesting a possible anticipation of the growing season.

While yearly precipitation trends are overall negative, the quarterly analysis shows that a significant decrease of precipitation in the first and second quarters occurred from 1959 to 2017, confirming that precipitation trends in NW Italy depend on the

considered temporal aggregation (Brunetti et al., 2006). The drainage trend analysis shows an overall reduction in the first and second quarters, with larger decreases in $AMJ$, but significant only for six watersheds, again in the western driest part of the region. The third and fourth quarters show slight positive non-significant trends.

A visual comparison between yearly and quarterly analysis is shown in Figure 7, where precipitation (top panel) and drainage (bottom panel) trends are displayed over the whole study area. This figure clearly shows again the spatial pattern of hydrological

changes, with negative trends in both precipitation plus snowmelt and discharge in spring. The dry western area is mainly represented by the Dora Riparia valley. Beside the spatial issue, the quarterly results show the importance of the time variability of precipitation. It confirm the importance of the computation at the daily scale (Healy, 2010). Masbruch et al. (2016) has shown the sensitivity of drainage to rainfall time variability. Finally, regarding actual evapotranspiration, Fatichi and Ivanov (2014) stressed the importance of short periods (order of hours or days) with high AET. Even if the present computations were

performed at the daily scale, this last point is out of the scope of the present work.

### 6.2 Mid-century projections of drainage

This section reports the results for the future projections of drainage. As already mentioned, this variable allows to quantify the groundwater resource availability by using only meteorological variables provided by the climate models. The simulations have been performed up to 2050 using the ensemble of simulations obtained with the RCM described in Section 5.1, at both

hydrological year and quarters scales.

#### 6.2.1 Hydrological year analysis

Table 5 shows the drainage trend slopes from 2018 up to 2050 for each catchment in the study area using climate projections under the RCP 4.5 and the RCP 8.5 scenarios. Bold font cells indicate trend values that are statistically significant. Drainage trends in the RCP 4.5 scenario are often negative but not statistically significant. These negative trends become stronger and

occasionally statistically significant in the more extreme RCP 8.5 scenario.

As a general statement, taking into account also the variability within the projection ensemble, the rather small values of drainage trends and their limited significance suggest that the Greater Turin metropolitan area will probably experience a yearly steady-state drainage situation till 2050, with total precipitation that shows a slight decrease. We can also notice a slight drainage increase with higher precipitation amounts, a decreasing trend with higher daily maximum temperature and, above

all, a strong interannual variability (see, as an example, the standard deviation evaluated for the detrended timeseries shown in Figure 8).

The projections of $P_q - AET$ (drainage) for two different cross-sections of the study area, the Dora Riparia station in Turin (catchment ID 66) and the Orco station in S. Benigno (catchment ID 109), are shown in Figure 8. Each row corresponds to

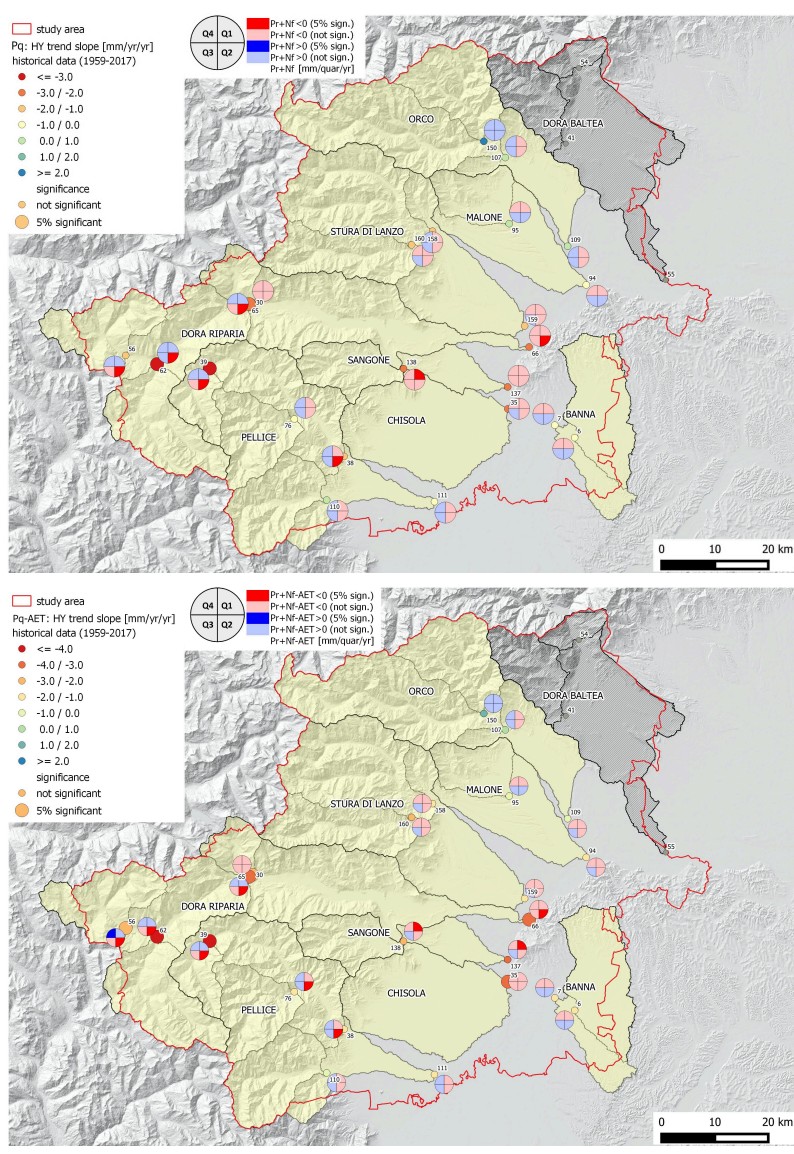

**Figure 7.** Trend slope at the catchment level for all study-area catchments. Top: $P_q$, bottom: drainage. Hydrological year (full dots) and quarterly trend values (four-segments circles) are jointly represented, trend significance is also indicated. When Quarters are considered, the precipitation is the sum of liquid precipitation $P_r$ and melted snow $N_f$. Topographic shading is based on DEM data from the Progetto Risknat - Base topografica transfrontaliera, ARPA Piemonte (http://webgis.arpa.piemonte.it/ags101free/rest/services/topografia_dati_di_base/Sfumo_Europa_WM/MapServer).




**Table 5.** Yearly trend slope of drainage projections. Bold font cells indicate trend significance.

| Primary catchments | Cross section ID number | HY [mm/yr/yr] | | | | | | | | | |
|---|---|---|---|---|---|---|---|---|---|---|---|
| | | CNRM-CM5 RCP4.5 | EC-Earth RCP4.5 | IPSL-CM5A-MR RCP4.5 | HadGEM2-ES RCP4.5 | MPI-ESM-LR RCP4.5 | CNRM-CM5 RCP8.5 | EC-Earth RCP8.5 | IPSL-CM5A-MR RCP8.5 | HadGEM2-ES RCP8.5 | MPI-ESM-LR RCP8.5 |
| | | $P_q$-AET | $P_q$-AET | $P_q$-AET | $P_q$-AET | $P_q$-AET | $P_q$-AET | $P_q$-AET | $P_q$-AET | $P_q$-AET | $P_q$-AET |
| Dora Riparia | 30 | 0.083 | -1.930 | -0.529 | -4.923 | 1.683 | 1.151 | 2.100 | -2.346 | -8.061 | 4.032 |
| | 56 | 1.327 | -0.672 | -2.467 | -1.632 | 2.236 | 2.201 | 0.887 | -4.431 | -3.308 | 4.265 |
| | 62 | 0.580 | -2.120 | -1.785 | -5.140 | 0.468 | 0.673 | 3.951 | -6.921 | -6.519 | 2.308 |
| | 65 | 0.829 | -1.657 | -2.159 | -4.077 | 1.353 | 0.884 | 2.404 | -5.270 | -5.698 | 3.322 |
| | 66 | 1.079 | -1.862 | -1.332 | -5.872 | 1.174 | 1.146 | 3.972 | -4.639 | -8.507 | 4.380 |
| Orco | 107 | 1.255 | -2.274 | -0.124 | -7.055 | 1.493 | 2.424 | 4.876 | -6.566 | -13.695 | 4.182 |
| | 109 | 1.526 | -2.441 | -0.326 | -7.649 | 1.666 | 2.502 | 5.053 | -6.299 | -15.089 | 4.904 |
| | 150 | 0.836 | -2.378 | -0.946 | -6.451 | 0.928 | 2.276 | 4.717 | -9.231 | -12.897 | 3.019 |
| Pellice | 38 | 0.630 | -3.057 | -3.848 | -7.876 | 0.741 | -0.081 | 5.533 | -8.578 | **-10.339** | 0.720 |
| | 39 | -0.603 | -2.427 | -3.715 | -5.666 | 0.367 | 0.302 | 3.388 | **-8.158** | **-7.451** | 0.172 |
| | 76 | 0.047 | -3.275 | -4.556 | -7.796 | 0.659 | -0.030 | 5.635 | **-10.842** | **-10.201** | -0.863 |
| | 110 | 1.170 | -3.376 | -4.615 | -10.036 | 0.593 | -1.113 | 7.959 | -12.009 | **-12.135** | -0.556 |
| | 111 | 1.416 | -3.239 | -3.981 | -8.800 | 0.838 | -0.511 | 6.765 | -9.143 | **-11.324** | 0.764 |
| Stura Lanzo | 158 | 1.009 | -1.783 | -0.168 | -9.843 | 1.344 | 2.705 | 6.199 | -5.487 | -14.937 | 5.970 |
| | 159 | 1.714 | -1.849 | -0.123 | -9.904 | 1.292 | 2.400 | 6.601 | -5.192 | -16.179 | 6.478 |
| | 160 | 0.778 | -1.509 | -0.591 | -9.629 | 0.647 | 3.032 | 6.789 | -5.257 | -14.074 | 6.726 |
| Banna | 6 | 0.308 | -3.243 | -2.756 | -2.519 | 0.084 | 1.300 | 3.120 | -5.330 | -7.308 | 1.681 |
| | 7 | 0.430 | -3.167 | -2.705 | -2.390 | 0.080 | 1.238 | 2.998 | -5.192 | -6.942 | 1.829 |
| Chisola | 35 | 2.814 | -2.791 | -2.395 | -6.854 | 0.599 | 2.143 | 7.679 | -5.060 | -11.663 | 6.284 |
| Malone | 94 | 2.849 | -2.861 | -0.894 | -9.472 | 1.588 | 2.053 | 5.872 | -4.591 | -20.158 | 7.182 |
| | 95 | 3.749 | -2.396 | 0.389 | -12.051 | 2.370 | 2.479 | 7.287 | -2.430 | -22.448 | 9.860 |
| Sangone | 137 | 2.372 | -2.294 | -1.222 | -8.119 | 0.682 | 1.125 | 7.281 | -4.689 | -12.594 | 5.846 |
| | 138 | 2.302 | -2.593 | -1.750 | -9.336 | 0.591 | 0.641 | 7.852 | -4.528 | -12.851 | 5.544 |





each of the five GCMs which drive the RCM; the results obtained with the RCP 4.5 (RCP 8.5) scenario are in red (green). The
slope of the regression line, the coefficient of determination $R^2$, the p-value and the standard deviation are also indicated. We
choose to show these two cases because the two catchments are among the largest ones (as from Table 4, 1243 km$^2$ and 829
km$^2$, respectively) and they represent, respectively, the drier (western) and the wetter (northern) parts of the region.

The overall results at yearly time scale, not shown for the sake of brevity, can be summarized as follows:

– $T_{max}$ shows positive trends, that are almost always significant for all model realizations and both scenarios. Also $T_{min}$
has similar trends;

– $P_q$ has either positive or negative trends, rarely significant;

– $AET$ has either positive and negative trends, rarely significant;

– slight positive drainage trends with higher precipitation amounts, negative trends with higher $T_{max}$ and high interannual
variability.

A large variability between the different projections was obtained, as in Crosbie et al. (2013) and in Persaud et al. (2020),
with a much less clear pattern in spatial variability if compared with the past data trends evaluations.

### 6.2.2  Quarterly analysis

Projections of drainage evaluated at the seasonal timescale show again a large variability within the ensemble of projections.
However, a general tendency to drainage increase in the first quarter and to drainage decrease in the third and fourth quarters,
$JAS$ and $OND$, emerges. Tables 6–9 show the drainage trend (expressed in mm/quarter/year) in the four Quarters. All models
and scenarios are indicated. Bold font cells indicate trend values that are statistically significant.

In the first quarter ($JFM$) the models mostly agree to indicate a drainage increase till 2050 over the whole study area, with
a wider agreement for RCP4.5 (4 out of 5 models concur in all catchments). In the second quarter ($AMJ$) in all river basins at
least 3 out of 5 models estimate a drainage decrease for the RCP 8.5 scenario, not for RCP4.5. In the third (3 out of 5 models)
and fourth (4 out of 5 models) quarters there is an overall tendency of drainage decrease in the whole study area for RCP4.5.

For the other meteoclimatic and water balance variables, at quarterly timescale we can report that (we omit the details for
brevity):

– $T_{max}$ and $T_{min}$ show positive trends in all quarters, almost always significant for all models and scenarios;

– in all quarters both rainfall plus snowmelt $P_q$ (including snowmelt) and rainfall ($P_r$) have both positive and negative
trends, rarely significant. Rainfall has almost always a weak positive trend in the first quarter, rarely significant. Snowfall
shows broad negative trends, almost always significant;

– $AET$ shows broad positive trends in the first and second quarters, despite some differences between the models. In the
third quarter an overall decreasing tendency can be observed, while in the fourth quarter trend slopes have values close
to zero.



**Figure 8.** Yearly drainage projections (RCP4.5 scenario in red and RCP 8.5 scenario in green) for two different cross-sections (Dora Riparia in Torino and Orco in San Benigno) of the study area. Each row corresponds to the results obtained with one GCM driving the RCA4 RCM.





**Table 6.** Trend slope of drainage projections in $JFM$ quarter. Bold font cells indicate trend significance.

| Primary catchments | Cross section ID number | JFM [mm/quar/yr] | | | | | | | | | |
|---|---|---|---|---|---|---|---|---|---|---|---|
| | | CNRM-CM5 RCP4.5 $P_T+N_T$-AET | EC-Earth RCP4.5 $P_T+N_T$-AET | IPSL-CM5A-MR RCP4.5 $P_T+N_T$-AET | HadGEM2-ES RCP4.5 $P_T+N_T$-AET | MPI-ESM-LR RCP4.5 $P_T+N_T$-AET | CNRM-CM5 RCP8.5 $P_T+N_T$-AET | EC-Earth RCP8.5 $P_T+N_T$-AET | IPSL-CM5A-MR RCP8.5 $P_T+N_T$-AET | HadGEM2-ES RCP8.5 $P_T+N_T$-AET | MPI-ESM-LR RCP8.5 $P_T+N_T$-AET |
| Dora Riparia | 30 | 0.983 | 0.611 | 1.764 | -5.140 | 0.726 | 1.463 | 1.599 | 1.495 | 0.408 | 1.140 |
| | 56 | 0.809 | -0.033 | -0.458 | -1.262 | 0.542 | 2.188 | -0.414 | -0.630 | 0.923 | 1.652 |
| | 62 | 1.169 | 0.848 | 1.270 | -3.522 | 0.014 | 1.263 | 1.425 | -0.078 | 0.112 | 0.796 |
| | 65 | 1.026 | 0.573 | 0.490 | -3.173 | 0.244 | 1.528 | 0.829 | -0.012 | 0.340 | 1.125 |
| | 66 | 1.289 | 0.998 | 1.938 | -5.179 | 0.031 | 1.649 | 1.827 | 0.703 | -0.192 | 1.343 |
| Orco | 107 | 1.193 | 1.106 | 3.711 | -7.180 | 0.708 | 1.798 | 1.689 | 0.779 | -0.789 | 1.233 |
| | 109 | 1.175 | 1.421 | 4.201 | -7.918 | 0.693 | 1.890 | 1.865 | 0.984 | -1.266 | 1.395 |
| | 150 | 0.889 | 1.048 | 3.346 | -6.362 | 0.257 | 1.333 | 1.447 | 0.190 | -0.846 | 0.490 |
| Pellice | 38 | 1.597 | 1.389 | 1.671 | -6.295 | -0.622 | 1.144 | 2.373 | 0.283 | -0.567 | 0.095 |
| | 39 | 0.959 | 1.026 | 0.865 | -3.954 | -0.284 | 0.930 | 1.190 | 0.133 | -0.291 | 0.233 |
| | 76 | 1.617 | 1.367 | 1.412 | **-5.893** | -0.560 | 0.994 | 2.172 | -0.005 | -0.407 | -0.266 |
| | 110 | 2.430 | 1.336 | 1.437 | -6.579 | -0.801 | 1.136 | 3.067 | -0.231 | -0.905 | -0.144 |
| | 111 | 2.002 | 1.558 | 1.893 | -6.897 | -0.902 | 1.236 | 2.930 | 0.072 | -0.903 | 0.004 |
| Stura Lanzo | 158 | 1.807 | 1.272 | 3.992 | -8.621 | 0.236 | 1.972 | 2.775 | 1.867 | -1.270 | 1.849 |
| | 159 | 1.784 | 1.695 | 4.615 | -8.871 | -0.147 | 2.154 | 3.044 | 1.795 | -2.097 | 1.930 |
| | 160 | 1.876 | 1.335 | 3.779 | -8.231 | -0.329 | 1.881 | 3.055 | 2.257 | -1.463 | 1.942 |
| Banna | 6 | -0.195 | 1.404 | 2.291 | -2.233 | -0.025 | 0.942 | 0.902 | -0.504 | -0.176 | -0.101 |
| | 7 | -0.219 | 1.451 | 2.228 | -2.251 | -0.087 | 0.979 | 0.856 | -0.519 | -0.192 | -0.080 |
| Chisola | 35 | 1.367 | 2.355 | 3.357 | -6.097 | -1.483 | 1.956 | 2.965 | 0.092 | -1.866 | 1.343 |
| Malone | 94 | 1.236 | 2.416 | 6.111 | -10.098 | 0.059 | 2.411 | 2.869 | 1.914 | -3.697 | 1.932 |
| | 95 | 1.911 | 2.687 | 6.825 | -13.023 | 0.256 | 3.129 | 3.160 | 3.170 | -4.505 | 3.316 |
| Sangone | 137 | 1.638 | 1.986 | 3.711 | -7.439 | -0.944 | 1.898 | 3.169 | 0.579 | -1.576 | 1.578 |
| | 138 | 1.992 | 1.758 | 3.359 | -8.504 | -0.869 | 1.753 | 3.425 | 0.775 | -1.276 | 1.547 |




**Table 7.** Trend slope of drainage projections in $AMJ$ quarter. Bold font cells indicate trend significance.

| Primary catchments | Cross section ID number | CNRM-CM5 RCP4.5 $P_r+N_r{^\wedge}AET$ | EC-Earth RCP4.5 $P_r+N_r{^\wedge}AET$ | IPSL-CM5A-MR RCP4.5 $P_r+N_r{^\wedge}AET$ | HadGEM2-ES RCP4.5 $P_r+N_r{^\wedge}AET$ | MPI-ESM-LR RCP4.5 $P_r+N_r{^\wedge}AET$ | CNRM-CM5 RCP8.5 $P_r+N_r{^\wedge}AET$ | EC-Earth RCP8.5 $P_r+N_r{^\wedge}AET$ | IPSL-CM5A-MR RCP8.5 $P_r+N_r{^\wedge}AET$ | HadGEM2-ES RCP8.5 $P_r+N_r{^\wedge}AET$ | MPI-ESM-LR RCP8.5 $P_r+N_r{^\wedge}AET$ |
|---|---|---|---|---|---|---|---|---|---|---|---|
| Dora Riparia | 30 | -1.647 | 0.417 | 0.573 | -0.053 | 0.591 | -2.453 | -0.398 | -2.070 | **-4.274** | 2.112 |
|  | 56 | -0.325 | 0.707 | -1.083 | 0.321 | 0.825 | -1.520 | -0.677 | -2.478 | **-2.645** | 1.616 |
|  | 62 | -1.606 | 0.863 | -0.327 | -0.592 | 0.039 | -1.516 | 2.217 | -4.091 | -3.454 | -0.241 |
|  | 65 | -1.222 | 0.823 | -0.398 | -0.123 | 0.459 | -1.739 | 0.755 | -3.256 | **-3.334** | 0.942 |
|  | 66 | -1.555 | 0.711 | 0.129 | 0.092 | 0.490 | -1.997 | 1.052 | -2.430 | **-4.386** | 1.782 |
| Orco | 107 | -1.370 | 0.740 | -0.495 | 1.029 | 0.229 | -2.759 | 0.788 | -2.115 | **-6.440** | 1.999 |
|  | 109 | -1.471 | 0.537 | -0.583 | 1.483 | 0.470 | -2.867 | 0.796 | -1.685 | **-6.636** | 2.349 |
|  | 150 | -1.164 | 0.945 | -0.364 | 1.527 | -0.028 | -2.365 | 1.176 | -2.609 | **-5.882** | 1.531 |
| Pellice | 38 | -1.807 | 0.703 | -0.062 | -0.253 | 0.000 | -1.879 | 2.376 | -3.676 | **-5.200** | 0.209 |
|  | 39 | -2.008 | 0.767 | -0.727 | -0.427 | -0.568 | -1.494 | 2.215 | -3.998 | -3.413 | -1.202 |
|  | 76 | -1.952 | 0.689 | -0.526 | -0.441 | -0.450 | -1.700 | 2.920 | -4.488 | -5.049 | -1.115 |
|  | 110 | -1.712 | 0.427 | -0.290 | -0.935 | 0.034 | -1.847 | 3.935 | -4.950 | **-6.468** | -0.818 |
|  | 111 | -1.644 | 0.457 | -0.094 | -0.239 | 0.362 | -1.919 | 2.833 | -3.673 | **-5.733** | 0.326 |
| Stura Lanzo | 158 | -2.435 | 1.099 | 0.099 | 0.022 | 0.179 | -2.897 | 1.355 | -2.377 | **-7.080** | 2.614 |
|  | 159 | -2.240 | 0.899 | 0.127 | 0.692 | 0.590 | -2.802 | 1.570 | -1.621 | **-7.177** | 2.907 |
|  | 160 | -2.681 | 1.168 | 0.408 | 0.158 | 0.032 | -2.403 | 1.967 | -2.274 | **-6.601** | 2.766 |
| Banna | 6 | -0.395 | -0.996 | -1.859 | 1.006 | 0.389 | -0.411 | 0.281 | -0.334 | -2.879 | 1.505 |
|  | 7 | -0.275 | -1.000 | -1.745 | 1.133 | 0.415 | -0.363 | 0.207 | -0.367 | -2.776 | 1.571 |
| Chisola | 35 | -0.601 | -0.150 | -0.238 | 1.183 | 1.693 | -1.030 | 2.304 | -0.212 | **-5.155** | 3.442 |
| Malone | 94 | -1.624 | -0.063 | -0.813 | 3.083 | 1.223 | -3.252 | 0.960 | 0.108 | **-7.564** | 3.623 |
|  | 95 | -1.870 | 0.280 | -0.165 | 3.640 | 1.464 | -4.013 | 1.495 | 0.451 | **-8.941** | 4.702 |
| Sangone | 137 | -1.239 | 0.454 | 0.571 | 0.694 | 1.119 | -1.791 | 2.195 | -0.647 | **-5.883** | 3.103 |
|  | 138 | -1.534 | 0.748 | 0.909 | 0.195 | 0.884 | -2.053 | 2.387 | -1.052 | **-6.646** | 3.282 |





**Table 8.** Trend slope of drainage projections in $JAS$ quarter. Bold font cells indicate trend significance.

| Primary catchments | Cross section ID number | CNRM-CM5 RCP4.5 $P_T+N_T-AET$ | EC-Earth RCP4.5 $P_T+N_T-AET$ | IPSL-CM5A-MR RCP4.5 $P_T+N_T-AET$ | HadGEM2-ES RCP4.5 $P_T+N_T-AET$ | MPI-ESM-LR RCP4.5 $P_T+N_T-AET$ | CNRM-CM5 RCP8.5 $P_T+N_T-AET$ | EC-Earth RCP8.5 $P_T+N_T-AET$ | IPSL-CM5A-MR RCP8.5 $P_T+N_T-AET$ | HadGEM2-ES RCP8.5 $P_T+N_T-AET$ | MPI-ESM-LR RCP8.5 $P_T+N_T-AET$ |
|---|---|---|---|---|---|---|---|---|---|---|---|
| Dora Riparia | 30 | 0.553 | -1.293 | 0.006 | -1.054 | 0.715 | 0.673 | 1.105 | -0.401 | 0.442 | -0.175 |
| | 56 | 0.758 | -0.589 | -0.105 | -0.826 | 1.107 | 0.555 | 1.351 | -0.742 | 0.777 | 0.284 |
| | 62 | 0.694 | -1.787 | -0.566 | -1.492 | 0.448 | 1.835 | 1.009 | -1.632 | 0.087 | -0.225 |
| | 65 | 0.720 | -1.318 | -0.250 | -1.189 | 0.746 | 1.136 | 1.156 | -1.010 | 0.451 | -0.104 |
| | 66 | 0.758 | -1.416 | -0.354 | -1.299 | 0.729 | 1.135 | 1.178 | -1.177 | 0.451 | -0.225 |
| Orco | 107 | 0.810 | -1.657 | -0.505 | -1.327 | 1.055 | 1.282 | 1.997 | -2.587 | 0.192 | -0.567 |
| | 109 | 0.841 | -1.750 | -0.453 | -1.467 | 1.009 | 1.472 | **1.982** | -2.523 | 0.131 | -0.522 |
| | 150 | 1.005 | -1.837 | -1.055 | -1.423 | 1.263 | 1.484 | **2.161** | **-3.826** | 0.054 | -0.643 |
| Pellice | 38 | 0.640 | -1.941 | -0.988 | -1.446 | 1.062 | 1.226 | 1.339 | -2.997 | -0.151 | -0.726 |
| | 39 | 0.510 | -1.764 | -1.055 | -1.320 | 1.003 | 1.602 | 1.172 | -2.635 | -0.384 | -0.491 |
| | 76 | 0.522 | -2.157 | -1.267 | -1.334 | 1.226 | 1.484 | 1.350 | **-4.061** | -0.500 | -0.794 |
| | 110 | 0.566 | -2.071 | -1.328 | -1.571 | 0.478 | 1.283 | 1.484 | **-4.728** | -0.431 | -1.206 |
| | 111 | 0.661 | -1.988 | -0.942 | -1.486 | 0.763 | 1.241 | 1.393 | **-3.313** | -0.134 | -0.781 |
| Stura Lanzo | 158 | 1.006 | -1.475 | -0.614 | -1.199 | 1.265 | 1.605 | **1.741** | -2.194 | 0.420 | -0.586 |
| | 159 | 0.991 | -1.522 | -0.534 | -1.372 | 0.956 | 1.676 | **1.671** | -2.061 | 0.402 | -0.487 |
| | 160 | 1.097 | -1.430 | -1.157 | -1.324 | 1.335 | 1.921 | 1.568 | -2.556 | 0.484 | -0.575 |
| Banna | 6 | 0.598 | -2.456 | -0.351 | -1.530 | 0.425 | 0.859 | 1.212 | -1.880 | -0.202 | -0.488 |
| | 7 | 0.568 | -2.424 | -0.368 | -1.492 | 0.383 | 0.856 | 1.228 | -1.769 | -0.057 | -0.469 |
| Chisola | 35 | 1.015 | **-2.203** | -0.575 | -1.833 | 0.032 | 1.681 | 1.434 | -1.998 | 0.041 | 0.174 |
| Malone | 94 | 0.932 | -1.909 | -0.241 | **-1.793** | 0.609 | 2.050 | 1.715 | -1.895 | 0.112 | -0.311 |
| | 95 | 1.050 | -1.595 | -0.118 | -1.677 | 0.842 | 2.172 | **2.094** | -1.585 | 0.730 | -0.577 |
| Sangone | 137 | 0.851 | -1.623 | -0.412 | -1.630 | 0.325 | 1.055 | **1.273** | -1.729 | 0.333 | -0.103 |
| | 138 | 0.858 | -1.623 | -0.532 | -1.543 | 0.593 | 0.802 | **1.382** | -1.912 | 0.520 | -0.399 |




**Table 9.** Trend slope of drainage projections in $OND$ quarter. Bold font cells indicate trend significance.

| Primary catchments | Cross section ID number | OND [mm/quar/yr] | | | | | | | | | |
|---|---|---|---|---|---|---|---|---|---|---|---|
| | | CNRM-CM5 RCP4.5 $P_r+N_r$-AET | EC-Earth RCP4.5 $P_r+N_r$-AET | IPSL-CM5A-MR RCP4.5 $P_r+N_r$-AET | HadGEM2-ES RCP4.5 $P_r+N_r$-AET | MPI-ESM-LR RCP4.5 $P_r+N_r$-AET | CNRM-CM5 RCP8.5 $P_r+N_r$-AET | EC-Earth RCP8.5 $P_r+N_r$-AET | IPSL-CM5A-MR RCP8.5 $P_r+N_r$-AET | HadGEM2-ES RCP8.5 $P_r+N_r$-AET | MPI-ESM-LR RCP8.5 $P_r+N_r$-AET |
| Dora Riparia | 30 | 1.022 | -1.391 | -1.621 | -0.095 | -0.860 | 0.493 | 0.332 | 0.012 | -3.665 | 1.741 |
| | 56 | 0.109 | -0.468 | 0.229 | -0.492 | -0.725 | 0.064 | 1.323 | 0.361 | -1.836 | 0.593 |
| | 62 | 1.223 | -1.445 | -1.175 | -0.422 | -0.603 | -1.404 | -0.195 | 0.352 | -2.297 | 1.316 |
| | 65 | 0.851 | -1.278 | -0.933 | -0.506 | -0.682 | -0.727 | 0.203 | 0.314 | -2.289 | 1.067 |
| | 66 | 1.271 | -1.697 | -1.856 | -0.687 | -0.893 | -0.374 | 0.429 | -0.234 | -3.315 | 1.584 |
| Orco | 107 | 1.529 | -1.634 | -1.695 | -1.162 | -1.370 | 1.103 | 1.353 | -0.630 | -5.494 | 1.844 |
| | 109 | 1.867 | -1.895 | -2.242 | -1.442 | -1.534 | 1.084 | 1.154 | -1.122 | -5.848 | 2.134 |
| | 150 | 0.974 | -1.525 | -1.963 | -1.489 | -1.256 | 0.721 | 1.026 | -0.844 | -5.186 | 1.200 |
| Pellice | 38 | 1.329 | -2.669 | -3.278 | -0.884 | -0.670 | -1.188 | -0.016 | 0.180 | -3.142 | 0.601 |
| | 39 | 1.050 | -2.058 | -1.915 | -0.601 | -0.493 | -1.417 | -0.664 | 0.226 | -2.366 | 0.556 |
| | 76 | 1.268 | -2.635 | -3.151 | -0.873 | -0.474 | -1.452 | -0.198 | 0.212 | -3.060 | 0.301 |
| | 110 | 1.500 | -2.544 | -3.411 | -1.608 | -0.443 | -1.799 | -0.136 | 0.583 | -2.986 | 0.667 |
| | 111 | 1.582 | -2.674 | -3.606 | -1.224 | -0.602 | -1.418 | 0.035 | 0.248 | -3.213 | 0.733 |
| Stura Lanzo | 158 | 1.899 | -2.315 | -2.315 | -1.706 | -1.684 | 0.789 | 1.067 | -0.865 | -5.664 | 2.501 |
| | 159 | 2.262 | -2.480 | -2.941 | -2.015 | -1.633 | 0.368 | 0.869 | -1.517 | -5.640 | 2.529 |
| | 160 | 1.797 | -2.270 | -2.362 | -1.766 | -1.702 | 0.335 | 0.908 | -0.929 | -5.070 | 2.756 |
| Banna | 6 | 0.645 | -1.124 | -1.490 | -0.440 | -0.960 | -0.332 | 1.002 | -1.147 | -2.874 | 1.154 |
| | 7 | 0.671 | -1.147 | -1.481 | -0.433 | -0.912 | -0.442 | 0.971 | -1.042 | -2.808 | 1.231 |
| Chisola | 35 | 1.586 | -2.148 | -3.213 | -1.225 | -0.802 | -0.699 | 1.283 | -1.387 | -3.059 | 1.906 |
| Malone | 94 | 3.029 | -2.792 | -4.375 | -2.490 | -1.985 | 0.272 | 0.446 | -3.106 | -6.535 | 2.635 |
| | 95 | 3.586 | -3.276 | -4.607 | -3.092 | -2.428 | 0.525 | 0.765 | -3.039 | -7.291 | 3.429 |
| Sangone | 137 | 1.737 | -2.526 | -3.595 | -1.101 | -1.078 | -0.389 | 1.032 | -1.189 | -3.973 | 1.871 |
| | 138 | 1.654 | -2.842 | -4.002 | -0.998 | -1.287 | -0.282 | 1.098 | -0.418 | -4.089 | 1.742 |





As reported by Stoll et al. (2011) in their recharge projections about a catchment in northern Switzerland, the precipitation time variability plays the major role. Konapala et al. (2020) indicate changes in long-term drainage, but they recall the limitations in the ability of current generation coupled climate models to capture the key drivers of persistent weather extreme.

## 7  Conclusions

Assessing the impacts of climate change on groundwater resources represents a priority in water management, besides being
an important scientific challenge. In this study we focused on an area where the aquifer providing water to about 2.3 million people is catched in watersheds characterized by a very large spatial variability of precipitation, due to the proximity of high mountains and of the sea. In this work we assessed the trends of precipitation, temperature and actual evapotranspiration in order to estimate the trend of drainage as a proxy of the water available for drinking purposes. The analyses have been performed both at the hydrological year and at the quarterly timescales. We analysed past and future conditions, using a
historical climate dataset providing temperature and precipitation data for the period 1959–2017 and future projections from a multi-member ensemble of a regional climate model up to 2050, in order to be compliant with the time frame of water management objectives. Our analysis revealed a very strong interannual variability of drainage in the historical period, as well as remarkable spatial differences, i.e., across the different watersheds. We found that the driest part of the region (the Dora Riparia Valley) shows significant negative drainage trends both yearly and in the spring quarter. The increasing trend of actual
evapotranspiration is typical of colder and moister areas, like England (Blyth et al., 2018). The combination of it with a very variable precipitation input has to be monitored in the future for its potential effects on drainage. As for the future projections, there is a large inter-model variability, as in Crosbie et al. (2013) and in Persaud et al. (2020), of all hydrological variables, which is evident for all catchments, and almost no significant trends. This last result is in line with Gudmundsson et al. (2017) for this part of Europe. Runoff was not considered in this study, but following Kumar et al. (2016) it could be useful to quantify
its variability with climate change.

This study constitutes a knowledge basis which helps for a better informed management, infrastructural and supply decisions for the study area considered, and our methodology could be extended also to other regions. This research is the result of a shared work developed together with a drinking water authority and academic/research institutions, in a general context which stimulated and supported a participatory planning that is scientifically grounded. Our results will be integrated in the
definition and refinement of study-area long-term guidelines and strategic development for groundwater resources protection and infrastructural provision and planning as a part of a wider effort to strengthen good water resource management and governance.

*Data availability.*  The datasets presented in this study can be obtained upon request to the corresponding author.



*Author contributions.* All the authors conceived the study. EP and JvH provided and pre-processed the meteoclimatic data. GM, GV, MP,
DC, DG, IB and SF performed and analysed the hydrological simulations and prepared some of the figures. EB and SF analysed the results.
EB and IB prepared the figures and EB all the tables. EB, EP, JvH, AP and SF wrote the paper with support from all the authors.

*Competing interests.* The authors declare that they have no conflict of interest

*Acknowledgements.* This work has been funded by Società Metropolitana Acque Torino S.p.A. Research has been supported in part by
"Dipartimento di Eccellenza" DIST department funds and "PRIN MIUR 2017SL7ABC_005 WATZON Project" and by the Project of Na-
tional Interest NextData of the MIUR (Italian Ministry for Education, University and Research). The authors would like to thank the Risk
Responsible Resilience Interdepartmental Centre (R3C) DIST- PoliTO for valuable collaboration and ARPA Piemonte for kind support. The
observational datasets used in this study are all freely available. Post-processed data are reproducible following the detailed descriptions of
Section 2 and 4 and available upon request to the authors.





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
