# Peer review of "Aquifer recharge in the Piedmont Alpine zone: Historical trends and future scenarios"

_Hydrology and Earth System Sciences, 2020_

## Referee Comment (RC2)

[referee-annotated manuscript omitted]

---

## Author Comment (AC1)

**Reference: hess-2020-501**

"Aquifer recharge in the Piedmont Alpine zone: Historical trends and future scenarios"

**Reviewer #1**

1) *The manuscript sent by Brussolo et al. provides a study and information of water potential for drinking water supply in the catchment area that belongs nearly 2.3 million inhabitants. The past, present and future state of the water potential is studied. Brussolo et al. provides a methodology that accounts for precipitation, temperature, snow melt and actual evapotranspiration. Climate change scenarios are predicted by RCM (RCA4). I think this study serves the local policy very well and will benefits the local measurements of future usage of water consumption as they also mention in conclusions "This study constitutes a knowledge basis which helps for a better informed management, infrastructural and supply decisions for the study area considered…" Brussolo et al concluded that "…our methodology could be extended also to other regions".*

We thank the reviewer for this positive feedback.

2) *The methodology presented here is not very novel and I encourage the authors to look much more detail in the scientific literature about the models that have been used in the prediction of water potential in local, regional and even continental scale. Authors do not provide any background about previous studies of methods that have been used in the past 20 years. For example page 2, 30, they list impacts of CC impacts on hydrological cycle but do not provide any references of previous studies.*

We revised the Introduction to include more references providing a background about previous studies (Smerdon 2017; Epting et al., 2021; Li et al., 2015, Taylor et al. 2013) which justify the use of precipitation as the largest source of recharge variability, the importance of the spatial and temporal variability in recharge-related studies and illustrate how groundwater recharge projections are related to projections of precipitation, irrigation and snowpack  reduction.

3) *The main objective of the paper is to estimate future change and variability of precipitation, AET and drainage (recharge) locally. This kind of assessment is more or less day-to-day job nowadays which means that novelty with respect to scientific research should be emphasis more clearly in this study. If they like to present a novel method or approach, and use this region as an example, they need to clearly state how this method/approach improves the previous ones that have been used. If the goal is to assess water potential of this region, then they need to come up with the outcome that is not noted in previous studies (in similar environment) and clearly emphasis this outcome. Unfortunately I did not recognize this. I encourage authors to get back on board, and revise the paper in the sense that it could focus more on the methodology and clearly state the novelty of it, or if the focus is more on the region itself, the novelty of simulations results need clearly state and what are new findings that could attract readers, not only from this region but also elsewhere.*

We thank the reviewer for this comment that allows us to better explain the focus of the paper.
In fact, in our opinion this paper can be considered as a case-study of an interdisciplinary research approach, where scientists in atmospheric/climate research and  hydrologists  work together with agricultural and soil scientists and experts from a water utility  to quantify the role of groundwater in providing water for society. A local water utility (to which the first author is affiliated) provided funding for this study and participated in defining the goals of the research and in the development of the analysis. This paper shows the results of an approach really tailored to stakeholder needs, and the outcomes will drive local policies, helping to assess the water potential of the study area. This study provides the knowledge basis for the definition and refinement of the study-area long-term guidelines and strategic development for groundwater resources protection and infrastructural provision and planning.

In such a context, it is important to highlight that no similar "stakeholder-driven" research studies were already carried out in that area and this approach anticipates the recent EU Sustainable Taxonomy (REGULATION (EU) 2020/852) that clearly states that all the mitigation and adaptation measures undertaken by the different stakeholders should be quantified and driven by state-of-the art climate information and projections.

The estimation of future change and variability of precipitation and other water balance terms are currently provided by different climate services already developed by the scientific community, however these services require an appropriate engagement with users which is still lacking to a large part. Often the stakeholders are not considered as proactive actors but only as final users who download pre-computed decision-relevant scientific information in order to develop and apply adaptation or mitigation strategies. In such a framework, this paper provides an example of a shared and sinergical work. We think that this study is totally in-line with the official aim and scope of HESS, to serve both the hydrological science community as well as water engineers and water managers.

Moreover, the characteristics of spatial and temporal variability of aquifer recharge found in this paper can be of interest for other European areas, particularly in the Mediterranean area and in the Alps. The paper not only relies on future climate simulations to provide estimates of future aquifer recharge, but also analyses about 60 years of historical data related to the water balance terms. The dry east-west valley shows significant negative drainage trends both yearly and in the spring quarter, confirming the seasonal variations found by Epting et al. (2021), and giving interesting hints for many other similar valleys in the Alps. Actual evapotranspiration is shown to generally increase both in historical data and in future simulations, despite the role of regulation given by soil drying. Finally, as suggested by Taylor et al. (2013), the role of irrigated agriculture is considered. In such a context, irrigation, that significantly contributes in increasing actual evapotranspiration as a consequence of the air temperature increase, is simulated in a novel way. More specifically, the results of field studies with both surface and sprinkler irrigation methods were used, combining them with crop and irrigation regional databases (as described in the Sections "Study area and methodology" and "Input data of the soil water model" of the manuscript.)

All these considerations have been included in the revised paper extending the abstract, introduction, methodology and conclusions.

 4) *The results need (in both cases) to discuss in depth and compare with previous findings.*

Actually, our manuscript already discusses and compares exhaustively the results to previous findings, as also demonstrated by our extensive bibliography (which has been further extended following our reply #2 to this reviewer). If there are specific points which the reviewer thinks should be discussed in more depth, we will be glad to address them.

---

## Author Comment (AC2)

**Reference: hess-2020-501**

"Aquifer recharge in the Piedmont Alpine zone: Historical trends and future scenarios"

**Reviewer #2**

1) *the HESS-manuscript submission "Aquifer recharge in the Piedmont Alpine zone: Historical trends and future scenarios" by Brussolo et al. addresses the impacts of climate change on the components of the water balance in northwestern Italy. Overall, the manuscript is well written and structured.*

We thank the reviewer for this positive feedback.

2) *Generally, the paper could be shortened by omitting repetitions; Tables could be prepared as illustrations or moved to the supplementary information;*

We have welcomed the reviewer's suggestion and added a "supplementary Information" file where all the tables that summarize the main results discussed in the paper are included.

3) *Figures 1 & 2 could be merged.*

Figures 1 and 2 have been merged in the new paper version, as suggested by the reviewer. Thank you.

4) *At several locations (indicated in the annotated manuscript) specification would help.*

Details have been added at the points indicated by the reviewer in the annotated pdf..

5) *As the main topic is "aquifer recharge", the description of different groundwater recharge comes a little bit short. E.g. provide details on interaction processes of groundwater resources and surface waters (in- & exfitlration, etc.). Likewise, the whole link to groundwater (residual term of the water balance) and temperature imprinting is weak, resulting in a more qualitative assessment concerning the main topic "aquifer recharge".*

In order to address the reviewer comments and provide more details on the main topic of the paper, additional information was added in the introduction and in the methodology section (in the paragraph entitled "Water balance terms"). In particular, the Introduction has been extended as follows:

"Water is a crucial resource, intrinsically linked to society and culture development, food and energy security, well-being, environmental sustainability and poverty reduction. However, several factors, including urbanization, population growth, land use and soil consumption, industrial and agricultural development, endanger water resource sustainability in terms of availability, quality, management and demand (IPCC, 2014; WWAP, 2015). Groundwater resources represent about 97% of liquid freshwater resources on Earth (WHO, 2006; Healy, 2010) and play a key role in water supply and proper ecosystems preservation (WWAP, 2015). Groundwater resources help to maintain river discharges and, together with surface freshwaters, are accounted for in water budget considerations at the river basin scale (Rumsey et al., 2015). The hydrological connection between groundwater and surface water is primarily controlled by: (1) the driving force generated by the hydraulic gradient between groundwater and surface water, (2) the permeability degree of the aquifer in comparison to a streambed (i.e. different hydraulic conductivity) due to the geological context (Lasagna et al., 2016; Epting et al., 2018). Groundwater and surface water interaction is influenced by both local and regional regimes (Epting et al., 2018). Local interaction could be very complex and different methods

were developed to quantify this interaction in different locations (Bertrand et al., 2014; Kalbus et al., 2006). Groundwater resources are of utmost importance for their mitigation effects during dry periods and their reduction can impact the whole hydrological cycle. Groundwater is a fundamental natural resource that acts as a reservoir from which good quality water can be collected for drinking purposes, requiring few purifying treatments compared to surface water. Climate change influences several components of the water cycle, including groundwater resources, causing a lowering of piezometric levels due to discharge modifications as a result of snow retention reduction, changes in precipitation regimes and potential evapotranspiration that increases linearly according to temperature increase. In Alpine regions, shorter and thinner snow pack will decrease late spring flows, while the air temperature rise will increase stream flows in fall and winter due to trading of snowfall for rainfall (Confortola et al., 2013), leading to a shift of groundwater recharge from summer to winter, as evaluated by CH2018-Project-Team (2018) and Epting et al. (2021) in Switzerland.

Surface water and pollutants infiltration together with over-exploitation of wells can further deplete groundwater resources, triggering the competition between irrigation and potable uses. Though the degradation of water quality mostly depends on land use and saltwater intrusions into coastal groundwater (Jiménez Cisneros et al., 2014), climate change will affect, either directly or indirectly, also the quality of groundwater resources. Temperature impacts on biological, chemical and physical properties of groundwater resources (Epting et al., 2021), despite the increase of air temperature is not necessary directly correlated with groundwater temperature increase, depending on the intrinsic properties of aquifers (e.g. groundwater in plain aquifers responds to climate change with relatively long response times, in contrast to spring waters), on local and regional spatiotemporal scales and different anthropic inputs (Epting et al., 2021; Bastiancich et al., 2021). Moreover, the interaction between surface water and groundwater flow systems influences the water chemistry (Lasagna et al., 2016), since in areas where rainfall intensity is expected to increase pollutants will be increasingly washed from soils to water bodies (Parry et al., 2007). Finally, water-level changes are a key indicator that flow patterns are changing and that low-quality water may be mobilized (Moench et al., 2003)."

*Beside annotated manuscript here some more specific comments:*

1. *68: Recharge (for the sake of simplicity here defined as the difference between precipitation and actual evapotranspiration);even though maybe appropriate for the investigations this must be justified in more detail.*

As in our reply to reviewer 1, we revised the Introduction to include a background about previous studies, by adding new references (Smerdon 2017; Epting et al., 2021; Li et al., 2015, Taylor et al. 2013) which justify using precipitation as the largest source of recharge variability, the importance of the spatial and temporal variability in recharge-related studies and illustrate how groundwater recharge projections are related to precipitation, irrigation and snowpack reduction projections.

The temporal variations can be quantified for precipitation and actual evapotranspiration (heavily impacted by air temperature and, in this area, by irrigation). The other terms of the water balance equation are affected by higher uncertainty and their impact is smaller: soil water storage is small, river runoff has scattered measurements and complex process modelling, subsurface flow always shows large uncertainties associated with its estimation (Healy, 2010 p.35). Our paper addresses the calculation of the temporal variations of precipitation and evapotranspiration also in the historical data, confirming other studies focused on precipitation in the Italian territory (e.e. Libertino et al., 2019). These considerations are already discussed in the introduction of the original manuscript.

2. *96 – 103: move to methods section?*

Done.

3. *138: The difference between surface and subsurface catchments at least should be discussed to justify this assumption.*

Surface and subsurface catchments are assumed to be coincident as they are bounded by the border mountain divide, as considered in De Luca et al., 2020. We now specify this in the introduction and in the methodology.

4. *"not shown here": Maybe include in supplementary material?*

We have added this information in previous Table 4 of the paper, now Table S1 of the Supplementary Information file (which also includes Tables 5-9 of the previous paper version, now Tables S2-S6). Table S1 quantifies the interannual variability as the standard deviation of the detrended timeseries.

Finally, we thank the reviewer for all his/her remarks appearing in the annotated pdf which were considered in the revised manuscript.

---

## Author Response (AR1)

**Reference: hess-2020-501**

"Aquifer recharge in the Piedmont Alpine zone: Historical trends and future scenarios"

Dear Editor,

please find enclosed a point-by-point response to the comments and issues, with a detailed list of the changes which we applied. The original comments are shown in italics.

We have changed extensively the paper according to the comments raised by the reviewers and the Editor, in particular revising and extending, also including new references, the introduction, methodology and conclusions, integrating our results with information from the literature, underlying the scientific novelty of the paper and its relevance for an international audience.

We provide a copy of the manuscript with all changes tracked in red.

We thank the reviewers for their constructive comments which helped us to improve this work and we hope that it can now be considered for publication in HESS.

With best regards,

Elisa Palazzi

on behalf of the first author and all co-authors

**EDITOR COMMENTS:**

1) *Reviewer #1 thinks that this study serves the local policy very well but the methodology presented is not very novel. Reviewer #1 encourages the authors to better integrate the study results with the existing literature, with the aim of making it more relevant internationally and better bringing out its novelty. I concur with this assessment.*

*In the response the authors state that they are planning to revise the Introduction section, but I suggest they also, and even more so, integrate the methods and results with the existing literature. This is an internationally journal, so the papers not to be relevant for an international audience.*

*I would encourage the authors to revise their manuscript along the lines they proposed and, importantly, make the paper more relevant internationally as suggested by reviewer #1. Please list in detail the changes you have made, including in addressing the latter point.*

We have welcomed the suggestions by reviewer #1 and by the Editor and we further revised the paper extending the abstract, introduction, methodology, results and conclusions sections, and adding references to the relevant literature, as detailed below:

**INTRODUCTION**

The manuscript deals with the characterization of future groundwater resources availability in a study area located at the bridge between two climate hot-spot regions: the Alps and the Mediterranean area. In this area water resources are subject to multiple anthropogenic pressures and climate change is expected to exacerbate the competition between different users. [lines 65-68; lines 141-143]

The methodology developed in this study, a stakeholder-driven interdisciplinary research study, could be applied also in other areas of the world and our results will be useful to characterize future groundwater resources availability in other border areas between Mediterranean and continental climate. [lines 141-145]

The outcomes of this study are in-line with the existing literature, confirming the strong spatial variability of the precipitation field over northern Italy (Blöschl, 2019; Gudmundsson et al., 2017 and Libertino et al., 2019). We found that precipitation over Dora Riparia valley (one of the water catchments considered in our study) shows a decline, confirming a different behaviour compared to the rest of north-western Italy [lines 68-73].

In this study we addressed the following research questions, relevant also for applications in other regions                                of                                the                                world:
a) What is the role of AET and precipitation trends in determining significant trends in the water balance?
b) How different are water balance trends in three different portions of the mountain area, namely a drier west-east oriented area, a wetter area, and a mostly irrigated agricultural area?
c) To what extent are the spatial variability and trends observed in the past 60 years expected to undergo changes during the next 30 years? [lines 129-135]

All these research questions are now introduced in the introduction.

**METHODOLOGY**

The background about previous studies (Smerdon 2017; Epting et al., 2021; Li et al., 2015, Taylor et al. 2013) provided in the Introduction justifies the use of precipitation and of AET as the largest source of recharge variability. Also, the added literature highlights the importance of the spatial and temporal variability in recharge-related studies and illustrates how groundwater recharge projections are related to projections of precipitation, irrigation and snowpack  reduction [lines 92-110].

In such a framework, the widespread use of irrigation in most parts of the Po Valley plain needs to be correctly represented.  In managed agrosystems, changes in surface energy budgets are associated with enhanced soil moisture from irrigation (Taylor et al. (2013). Irrigation practices, especially for highly water-demanding crops such as maize, can increase the *AET* term of the water balance  [lines 240-243, lines 252-257]. These water-demanding crops are quite widespread in the study area and therefore a novel procedure reproducing agricultural irrigation techniques was implemented in the soil-water model. This module is based on previous research in three farms (Canone et al., 2015, 2016), reproducing farmers' decision rules, and tuned in order to obtain irrigation events similar to the observed ones [lines 252-257].

**RESULTS**

*Observed data (historical data from 1959 to 2017)*

The outcomes of this study are in-line with the existing literature, confirming the statistically significant positive trends of maximum temperature over the whole study area [lines 387-388, lines 392-393], and the strong spatial variability of the observed precipitation field over northern Italy (Blöschl, 2019; Gudmundsson et al., 2017 and Libertino et al., 2019) [lines 423-425].
Moreover, as reported in Blöschl (2019), it can be observed that the precipitation field over Dora Riparia valley (one of the water catchments considered in our study area) shows a significant negative trend, confirming a different behaviour compared to the rest of north-western Italian region. This is the driest part of the study area (i.e. the western part of the area), where the decrease

in precipitation combines with increasing evapotranspiration, thus leading to a significant negative drainage trend [lines 425-431].

The evaluation of temporal variability of precipitation is part of the international debate on Alpine precipitation variability as in Haslinger et al (2021) [lines 409-414].

In our paper, AET trends are positive over the whole study area, and statistically significant in 14 out of 23 catchments. This finding is in line with the results of Blyth et al. (2018), who estimated AET using a land surface model in Great Britain under similar conditions. That paper analysed a similar time period as in our paper, considering a region where precipitation is not out of phase with the vegetation growing season, similarly to our study area. However, the finding of an increase in AET is not obvious. For example, Pangle et al. (2014), using the data from a mesocosm experiment, found a decreasing trend of AET, highlighting that the hydrological response to climate warming can be attenuated where precipitation is out of phase with the vegetation growing season. In a more theoretical study, Fatichi and Ivanov (2014) found AET to be quite unaffected by the imposed climate fluctuations, using input data from four very different sites [lines 393-403].

Drainage shows negative but mostly non-significant trends, with significant decreasing trends only in the western part of the region (Dora Riparia catchment), where they combine with positive evapotranspiration trends. Increasing AET trends in the southern irrigated areas (a map representing the spatial distribution of annual AET has been added, in the revised paper see Figure 4) do not lead to a significant drainage decrease, because the precipitation field does not show a significant negative trend. Therefore, in our study area, precipitation plays a major role in affecting drainage trends [lines 423-432].

*Mid-century projections of drainage*

We used an ensemble of climate models to perform drainage projections, finding a large inter-model variability, as in Crosbie et al. (2013) and in Persaud et al. (2020). Moreover, a much less clear pattern in the spatial variability is found in the model projections compared with the historical data trend evaluations [lines 493-494].

Driven by the precipitation field, which does not show unique or significant trends, we find a slight drainage increase with higher precipitation amounts, decreasing trends with higher daily maximum temperature and, above all, a strong interannual variability [lines 467-471], as already reported by Stoll et al. (2011) in their recharge projections in a catchment in northern Switzerland [lines 513-514]. Changes in long-term drainage were already identified by Konapala et al. (2020), who highlighted the limitations in the ability of current generation coupled climate models to capture the key drivers of persistent weather extremes [lines 514-515].

**CONCLUSIONS**

Assessing the impacts of climate change on groundwater resources represents a priority in water management, besides being an important scientific challenge. In this study a stakeholder-driven research study was carried out to quantify the role of groundwater on an area characterized by a very large spatial variability of precipitation, due to the proximity of high mountains and of the sea [lines 517-522], located at the bridge between two climate hot-spot regions (the Alps and the Mediterranean area) and where multiple anthropogenic pressures act on groundwater resources [lines 555-558].

This study aims to support better informed management, infrastructural and supply decisions in the considered study area, with a methodology that could be extended also to other areas of the world

[lines 549-550]. Our results could help in better understanding future groundwater behaviour in other regions at the bridge between Central and Southern Europe [lines 559-562].

Regarding drainage, our analysis revealed a very strong interannual variability in the historical period, as well as remarkable geographical differences [lines 534-537] that are not evident in the model projections [lines 547-548]. In the analysis of historical observations, the western part of the study area (Dora Riparia catchment) shows significant negative trends, due to the combination of decreasing precipitation trends together with positive evapotranspiration trends [lines 540-542]. In this area a new drinking-water aqueduct (70 km long and 180 million cost) was built to provide water from an existing hydroelectric reservoir located at an elevation of 1600 m asl, preventing the continuous pumping from the Dora Riparia aquifer. A complementary hydraulic research was performed by the Polytechnic of Torino (Fellini et al., 2018) [lines 550-555].

Finally, the outcomes of this paper reinforce the findings of two different precipitation regimes over Europe, the Mediterranean and the continental one. Both can be found in our study area and the Dora Riparia valley seems to represent a transition between them [lines 561-563].

*2) Reviewer #2 has a number of more minor points which, I think, the authors addressed well in their response.*

Thank you.